# Nucleosome positioning shapes cryptic antisense transcription

Jian Yi Kok[1,2], Zachary H. Harvey[1], Elin Axelsson[1], Frédéric Berger[1]*

**1** Austrian Academy of Sciences, Gregor Mendel Institute, Vienna, Austria, **2** Vienna BioCenter PhD Program, Doctoral School of the University of Vienna and Medical University of Vienna, Vienna, Austria

* frederic.berger@gmi.oeaw.ac.at

## Abstract

Maintaining transcriptional fidelity is essential for precise gene regulation and genome stability. Despite this, cryptic antisense transcription, occurring opposite to canonical coding sequences, is a pervasive feature across all domains of life. How such potentially harmful cryptic sites are regulated remains incompletely understood. Here, we show that nucleosome arrays within gene bodies play a key role in suppressing cryptic transcription. Using the fission yeast *Schizosaccharomyces pombe* as a model, we demonstrate that the CHD-family chromatin remodeler Hrp3 coordinates with the transcription elongation machinery, via the transcriptional regulator Prf1/RTF1, to position nucleosomes at sites of cryptic transcription initiation within gene bodies. In the absence of Hrp3, AT-rich sequences within gene bodies lose nucleosome occupancy, exposing promoter-like sequences that drive cryptic initiation. While cryptic transcription is generally detrimental, we identify a subset of antisense transcripts that encode critical meiotic genes, suggesting that cryptic transcription can also serve as a source of regulatory innovation. These findings define an elongation-coupled chromatin pathway that preserves transcriptional fidelity and reveal how nucleosome remodeling shapes antisense transcription, cellular homeostasis, and adaptive potential.

## Author summary

Cells must control where transcription begins so that genes are expressed accurately. Yet RNA polymerase can also start from hidden sites within genes and produce antisense RNAs from the opposite DNA strand. Antisense transcription is common across species, but when it increases too much, it can disrupt gene regulation and reduce cellular fitness. How cells prevent inappropriate antisense transcription is still not fully understood. In this study, we used fission yeast to test how nucleosome positioning helps maintain transcriptional fidelity. We show that the chromatin remodeler Hrp3 suppresses widespread antisense transcription by positioning nucleosomes along gene bodies to cover promoter-like DNA

**Data availability statement:** Datasets from the analyses are provided as supplementary data files. The annotated list of antisense transcripts identified in this study are available as a GFF3 file at https://github.com/Gregor-Mendel-Institute/kok_2025/tree/main/files/output/annos. Code used for analysing the data and generating figures are also available (https://doi.org/10.5281/zenodo.18632984). NGS data are deposited in GEO under accession numbers: PRO-seq (GSE302374), MNase-seq (GSE302388), mRNA-seq (GSE302386), and ChIP-seq (GSE302387). All other raw and processed data are available in the manuscript and its supplementary information.

**Funding:** This work was supported by core funding from the Gregor Mendel Institute, Austrian Academy of Sciences (ÖAW), and the following grants from the Austrian Research Fund (FWF) (TAI304, P32054, P36231, PAT1104523, and PAT6138924 to FB; ESP213 to ZHH). ZHH was further supported by an EMBO Postdoctoral Fellowship (ALTF169-2020). The funders had no role in study design, data collection and analysis, decision to publish, or preparation of the manuscript.

**Competing interests:** The authors have declared that no competing interests exist.

sequences that would otherwise initiate transcription. When Hrp3 is disrupted, these regions lose nucleosome protection and antisense transcription increases genome wide. We also find that Hrp3 is targeted to actively transcribed genes through an interaction with the conserved transcription factor Prf1, linking chromatin remodeling to the transcription machinery. Finally, we identify a subset of antisense transcripts associated with meiotic genes, suggesting that cryptic transcription can be repurposed for regulated gene expression.

## Introduction

Antisense transcription, a pervasive phenomenon with deep evolutionary roots [1], occurs when RNA polymerase II (RNAPII) initiates transcription in the opposite direction of protein-coding genes [2]. Observed across prokaryotes [3–5], unicellular protists [6], plants [7] and animals [8,9], antisense transcription has been implicated in diverse regulatory functions, including the modulation of sense transcript levels [8,10–12], epigenetic regulation [13], and developmental processes [14–16]. However, when misregulated, it can lead to transcriptional interference [17] and genome instability [18], fueling the pathology of cancers and neurological disorders [19–22]. Despite its widespread occurrence, how cells balance the regulatory roles of antisense transcripts with the need to suppress their potentially harmful effects when misregulated remains poorly understood.

Chromatin plays an important part in transcription start site (TSS) selection, and in transcriptional regulation [23]. Nucleosomes—the basic units of chromatin composed of two subunits each of the histones H2A, H2B, H3 and H4 wrapped by ~147 bp of DNA—act as both physical barriers and dynamic regulators of transcription across the transcriptional unit [24–26]. Beyond the well-defined roles of the +1 nucleosome at the TSS [27–29], nucleosomes are also regularly positioned and spaced along gene bodies, forming stereotypical phased arrays [30,31]. Although nucleosome arrays have been proposed to play roles in transcriptional elongation and termination [32], preventing spurious transcription initiation [33–35] and safeguarding the genome [36], their role in regulating sense transcription is modest, and therefore the precise link between their placement and gene regulation has remained a subject of debate.

Mechanistic studies seeking to address the role of nucleosome arrays in transcription have predominantly focused on nucleosome remodelers, particularly the chromodomain helicase DNA-binding (CHD) and imitation switch (ISWI) families [37–42], which are broadly essential across eukaryotes and are known to play important roles both in maintaining chromatin structure and regulating transcription [41,43–45]. Whereas their role in this context has been proposed, in animals and plants, multiple paralogs of ISWI and CHD have evolved [46–48], making it challenging to perform efficient genetic interventions to study nucleosome phasing, and to deconvolve direct and indirect effects. Additionally, the INO80 remodeling complex also plays a role in nucleosome spacing [49–52], further complicating the analysis of phasing.

Beyond the canonical CHD, ISWI, and INO80 families, several additional nucleosome remodeler families have been implicated in chromatin remodeling–adjacent processes and transcriptional regulation. For example, Mot1/BTAF1 remodels TATA-binding protein (TBP)–DNA interactions to modulate promoter accessibility genome-wide [53,54], members of the FUN30 family have been reported to disassemble nucleosomes during transcriptional elongation [55], and the Swi2/Snf2 proteins SHPRH/Irc20 and Uls1/HLTF carry RING-type ubiquitin ligase activity alongside the conserved ATPase domain and have been linked to remodeling events during DNA repair [56,57]. Nonetheless, among remodelers, robust nucleosome-spacing activity that establishes regular gene-body phasing is well supported primarily for the CHD, ISWI, and INO80 families.

While many studies on nucleosome remodelers have focused on the action of their conserved ATPase domain which drives nucleosome remodeling [58–60], the contributions of accessory domains are less well defined. In light of the functional diversity observed among the non-canonical nucleosome remodeler families, these domains are increasingly recognized as key determinants of genomic targeting and cofactor engagement, coordinating interactions with histone modifications, DNA features, and partner proteins [41,42,61]. For example, the chromodomains in CHD-family remodelers bind methylated H3 tails [62] for proper localization, while the HAND-SANT-SLIDE domain in ISWI-family remodelers is critical for recognizing linker DNA, which helps space nucleosomes [63]. Despite this, the precise contributions of several other accessory domains to overall remodeler function and transcriptional regulation are still unclear.

Here, we leverage the fission yeast *Schizosaccharomyces pombe* as a simplified system to disentangle remodeler-specific contributions to nucleosome phasing and transcriptional regulation. In many eukaryotes, overlapping activities of CHD- and ISWI-family remodelers complicate causal attribution; even in budding yeast, Chd1 and Isw1/Isw2 act redundantly in phasing and termination [32,36,45], obscuring their individual contributions. By contrast, fission yeast lacks ISWI-family remodelers [64], placing greater reliance on CHD-family remodelers to maintain gene body nucleosome organization. Prior work established that loss of the CHD-family remodeler Hrp3 disrupts nucleosome positioning genome-wide and induces pervasive antisense transcription, whereas its paralog Hrp1 has comparatively modest effects [65–68]. Building on this foundation, we combine transcriptomics, chromatin profiling, and functional assays to show that Hrp3 establishes phased nucleosome arrays within gene bodies that suppress cryptic antisense initiation. We further demonstrate that the transcriptional regulator Prf1/RTF1 recruits the CHD remodeler Hrp3 to gene-body nucleosomes via the Hrp3 CHCT domain, mechanistically linking transcriptional elongation to Hrp3-dependent nucleosome phasing. Loss of Hrp3 or disruption of its interaction with Prf1 results in impaired nucleosome phasing, increased antisense transcription, and significant fitness defects under stress conditions. Additionally, we identify thousands of antisense transcripts, including functional protein-coding transcripts and transcripts with unknown coding potential. Many of these transcripts correspond to genes nested convergently within larger hosts, including a substantial cohort of meiotically upregulated genes. Their expression depends on Hrp3-mediated antisense repression during vegetative growth, and their misexpression is associated with meiotic defects. These findings provide new insights into the mechanisms by which nucleosome remodelers suppress cryptic transcription and coordinate expression of nested genes, offering a framework to better understand how their dysregulation may contribute to altered gene expression programs and disease.

## Results

### Hrp3 is the predominant CHD-family remodeler regulating nucleosome phasing and antisense transcription

To systematically investigate the functional impact of nucleosome remodelers on antisense transcription in the fission yeast *Schizosaccharomyces pombe*, we generated knockout mutants for eleven viable nucleosome remodelers. This included three groups of paralogs - the FUN30 group (*fft1*, *fft2*, and *fft3*), the CHD group (*hrp1* and *hrp3*), and the ULS1 group (*rrp1* and *rrp2*), as well as four additional remodelers: *irc20*, *mit1*, *snf22*, and *swr1*, the latter being involved in the exchange of the transcription-associated histone variant H2A.Z [69] (**Fig 1A**). Since remodelers are primarily associated

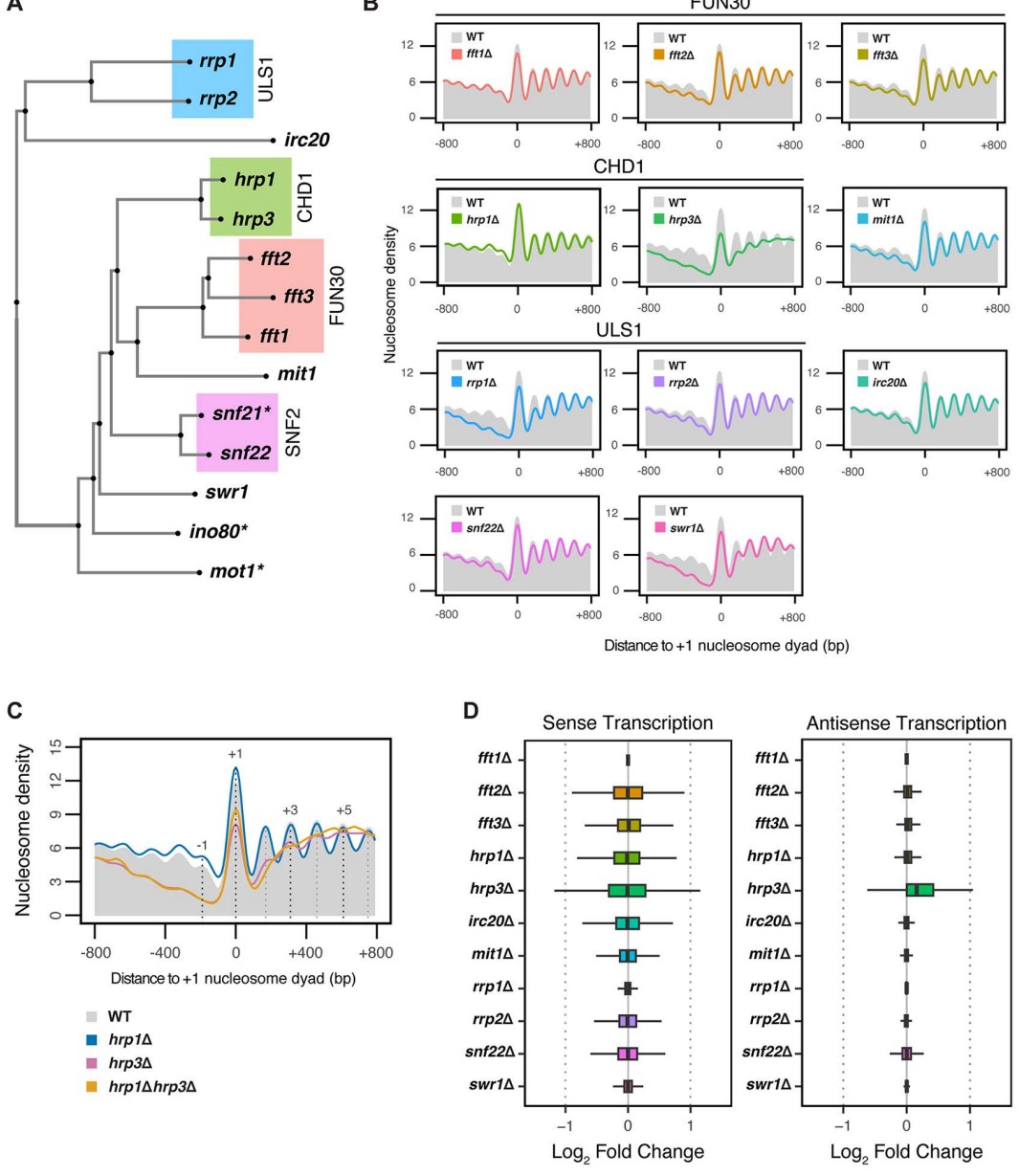

**Fig 1. The Impact of Nucleosome Remodelers on Nucleosome Phasing and Transcription.** (A) Unrooted tree generated from multiple sequence alignment of 14 nucleosome remodelers in *S. pombe*. Groupings of paralogs are indicated by coloring, with each group corresponding to the ortholog of a nucleosome remodeler in *S. cerevisiae*. Asterisks (*) indicate the remodelers that are essential in *S. pombe*. (B) Metaplots of normalized nucleosome density from MNase-seq centered on the +1 nucleosome dyad for *fft1Δ*, *fft2Δ*, and *fft3Δ*, *hrp1Δ*, *hrp3Δ*, *irc20Δ*, *mit1Δ*, *rrp1Δ*, *rrp2Δ*, *snf22Δ*, and *swr1Δ* mutants. Profiles include 800 bp upstream and downstream of the +1 nucleosome dyad for for all protein-coding genes. The WT profile is shown in solid grey. The plotted data represents the average signal from two biological replicates. (C) Metaplots of normalized nucleosome density from MNase-seq centered on the +1 nucleosome dyad for WT, *hrp1Δ*, *hrp3Δ*, and *hrp1Δhrp3Δ*. The WT nucleosome profile is shown in solid grey, with dotted lines indicating the positions of select nucleosome maxima (-1, +1, +3 and +5). Profiles include 800 bp upstream and downstream of the +1 nucleosome dyad for all protein-coding genes. The plotted data represents the average signal from two biological replicates. (D) Differential expression analysis of mRNA-seq data showing the boxplots of log2fold-change values of all *S. pombe* protein-coding genes in both sense and antisense orientations across *fft1Δ*, *fft2Δ*, and *fft3Δ*, *hrp1Δ*, *hrp3Δ*, *irc20Δ*, *mit1Δ*, *rrp1Δ*, *rrp2Δ*, *snf22Δ*, and *swr1Δ* mutants compared to WT. Data are calculated from three biological replicates.

with nucleosome dynamics, we assessed the impact of these mutations by first measuring genome-wide changes in nucleosome organization, followed by correlating these changes with sense and antisense transcription.

To examine nucleosome organization, we performed micrococcal nuclease digestion followed by sequencing (MNase-seq) to directly assess nucleosome phasing and profiled H2A.Z genomic distribution using chromatin immuno-precipitation followed by sequencing (ChIP-seq). Meta-profiles of nucleosome positioning centered on the +1 nucleosome indicated that WT and several remodeler mutants retained broadly ordered promoter-proximal nucleosome arrays, although mutants such as *swr1Δ* and *rrp1Δ* exhibited alterations in +1/+2 nucleosome occupancy and overall amplitude (**Fig 1B**). By contrast, *hrp3Δ* exhibited a marked loss of nucleosome periodicity beyond the +2 position (**Fig 1C**), consistent with prior reports implicating Hrp3 in gene-body phasing [65–68]. This pattern was recapitulated in *hrp1Δhrp3Δ*, whereas *hrp1Δ* alone showed minimal changes. Despite the strong impact of Hrp3 on gene body nucleosome phasing, we observed no effect on H2A.Z enrichment profile in *hrp3Δ* mutants (S1A and S1B in **S1 Fig**) [70,71], consistent with previous reports that H2A.Z deposition is exclusive to Swr1-C [69,72]. These results establish that Hrp3 is the primary CHD-family remodeler responsible for maintaining nucleosome organization over gene bodies.

We next investigated whether these changes in nucleosome organization were associated with changes in transcription. When summarized at the gene level across the genome, transcriptome analysis (mRNA-seq) revealed only mild changes in sense and antisense transcript levels across most mutants compared to WT, except for *hrp3Δ* (**Fig 1D**). Consistent with its strong impact on nucleosome phasing, *hrp3Δ* cells exhibited an increase in antisense transcript levels compared to WT (**Fig 1D**). In contrast, *hrp1Δ* showed only a slight increase, consistent with previous studies [65–67]. At the transcriptome-wide level, this modest effect is reflected by a small median shift with a narrow interquartile range (**Fig 1D**), indicating a limited effect size despite statistical significance in pairwise comparisons (**Fig 2B**). Furthermore, *hrp1Δhrp3Δ* double mutants had antisense transcript levels comparable to those in *hrp3Δ* alone (**Fig 2A and 2B**), suggesting limited or no functional redundancy between the two paralogs. Importantly, loss of one CHD paralog did not trigger compensatory upregulation of the other (S2A in **S2 Fig**). Additionally, no meaningful correlation between sense and antisense transcript levels was observed in both *hrp1Δhrp3Δ* and WT cells (S2B–S2F **in S2 Fig**), suggesting that antisense transcription in this double mutant minimally impacts sense transcription. Together, these genome-wide measurements of chromatin and RNA extend prior observations of antisense upregulation upon loss of CHD remodelers and demonstrate that CHD orthologs govern antisense transcription, with Hrp3 serving as the principal remodeler that suppresses antisense initiation.

## Hrp3 blocks cryptic promoters to suppress antisense transcription

To determine whether increased antisense transcription in *hrp3Δ* reflected elevated RNA polymerase II (RNAPII) activity, we performed Precision Run-On sequencing (PRO-seq) [73], a method that assesses nascent transcription. Consistent with the transcriptome data, *hrp3Δ* and *hrp1Δhrp3Δ* mutants displayed a pronounced increase in antisense PRO-seq signal compared to *hrp1Δ* across genes (**Fig 2C and 2D),** demonstrating that the antisense expression phenotype is driven by RNAPII-dependent transcription.

We hypothesized that the increased antisense expression in *hrp3Δ* was due to the shifts in the nucleosome profile that exposed regions normally occluded by nucleosomes, thereby creating permissive sites for RNAPII initiation. To test this, we first generated a high-resolution annotation of antisense transcripts in the CHD mutants. We used PacBio Iso-Seq [74] to produce a comprehensive long-read transcriptome for both WT and *hrp1Δhrp3Δ*. Through transcript clustering and collapsing, we identified 5,105 new antisense transcripts. This more than doubles the known number of antisense transcripts, bringing the new total to 9,016 transcripts. Using this expanded antisense transcript annotation, we found that 4,678 antisense transcripts were differentially expressed in *hrp1Δhrp3Δ* compared to WT, overlapping with 54.9% of all protein-coding genes in *S. pombe* (S3A in **S3 Fig**). We did not identify new sense transcripts in *hrp1Δhrp3Δ* compared to WT (see **S1 Note**).

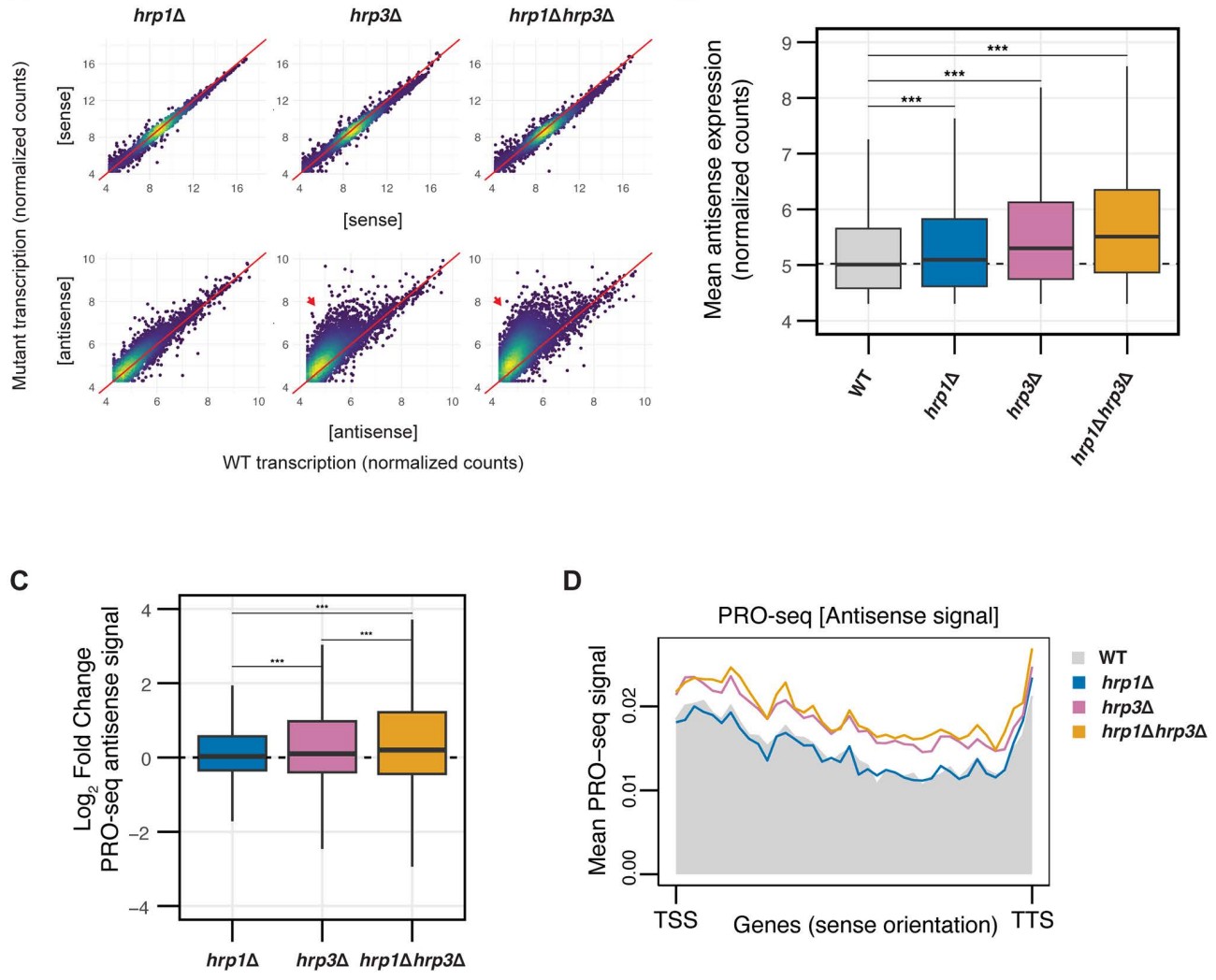

**Fig 2. Hrp3 is the Predominant CHD-family Remodeler Regulating Nucleosome Phasing and Antisense Transcription. (A)** Density-colored scatterplots comparing mutant (*hrp1Δ, hrp3Δ or hrp1Δhrp3Δ*) transcription to WT for sense transcripts (top row) and antisense transcripts (bottom row) for all protein-coding genes. Count data is derived from variance-stabilizing transformation (vst) of raw mRNA-seq counts. Red arrows indicate elevated antisense transcription in mutants relative to WT. **(B)** Boxplot of normalized antisense expression counts for all antisense genes for WT, *hrp1Δ, hrp3Δ* and *hrp1Δhrp3Δ*. The dotted line represents the median antisense expression in WT. Statistical analysis was performed using ANOVA with Tukey's Honestly Significant Difference (HSD) test. Asterisks indicate statistical significance: $p < 0.05$ (*), $p < 0.01$ (**), $p < 0.001$ (***), n.s. (not significant). **(C)** Boxplot of $\log_2$fold-change values of antisense expression measured by PRO-seq for *hrp1Δ, hrp3Δ* and *hrp1Δhrp3Δ* relative to WT. The dotted line represents a $\log_2$fold-change value of 0. Statistical analysis was performed using ANOVA with Tukey's HSD test. Asterisks indicate statistical significance: $p < 0.05$ (*), $p < 0.01$ (**), $p < 0.001$ (***), n.s. (not significant). Data are calculated from two biological replicates. **(D)** Metaplots of mean antisense PRO-seq signal for *hrp1Δ, hrp3Δ*, and *hrp1Δhrp3Δ* versus WT, plotted relative to the sense orientation of the transcription start sites (TSS) and transcription termination sites (TTS) of all genes.

We next validated our new antisense transcription annotations. As a control, aligning the transcription start sites (TSS) of protein coding genes with PRO-seq profiles revealed a peak corresponding to paused or recently initiated RNAPII within 200 bp downstream of the TSS on the sense strand (S3B in **S3 Fig**). Despite minor visual differences between datasets, mean sense transcription profiles are broadly comparable across strains, consistent with the minimal changes

observed by mRNA-seq (**Fig 2A**). Similarly, alignment of antisense transcription start sites (As-TSS) with PRO-seq data showed a peak immediately downstream of the As-TSS (**Fig 3A**), validating the high accuracy and precision of our antisense transcript annotation. Because PRO-seq measures nascent transcription initiation rather than steady-state RNA abundance, it is particularly sensitive to initiation-proximal antisense events. Consistent with the transcriptomics data, this PRO-seq peak was significantly stronger in *hrp3Δ* and *hrp1Δhrp3Δ* compared to *hrp1Δ* and WT (**Fig 3A and 3B**). We noticed that these antisense peaks coincided with nucleosome-shifted regions within gene bodies, which in WT were part of a regularly phased array (**Figs 3C**, S3C, and S3D in **S3 Fig**). When we stratified genes by *hrp1Δhrp3Δ* antisense expression, the WT showed increasing gene-body phasing with higher antisense quartiles, whereas *hrp1Δhrp3Δ* exhibited uniformly low amplitudes of nucleosome phasing (S3E and S3F in **S3 Fig**). This suggests that genes that exhibit high antisense expression in *hrp1Δhrp3Δ* are those that normally depend on robust gene-body phasing, linking remodeler-dependent phasing to suppression of cryptic antisense transcription.

To systematically investigate the relationship between nucleosome disruption and antisense initiation, we aligned the As-TSS with MNase-seq data. In all strains, the aggregate meta-profiles showed an apparently broad MNase-resistant feature at the As-TSS (**Fig 3D**). Heatmap visualization indicated that this broadening reflected positional heterogeneity of the nucleosome immediately adjacent to the As-TSS across genes, rather than an atypically large particle (S3G in **S3 Fig**). By contrast, downstream nucleosomes retained canonical spacing, with peak-to-peak distances of ~150 bp. In the mutants, this proximal nucleosome was preceded by a ~500 bp nucleosome-depleted region, forming a pattern that closely resembled the nucleosome organization at the sense TSS. In this region upstream of the As-TSS, *hrp3Δ* and *hrp1Δhrp3Δ* mutants exhibited significant loss of nucleosome phasing and occupancy (**Fig 3D and 3E**). In contrast, *hrp1Δ* mutants showed a smaller reduction in nucleosome occupancy in this region, accompanied by only a modest increase in antisense transcription. This suggests that the reduction in nucleosome occupancy and phasing in *hrp1Δ* alone is insufficient to induce the higher levels of antisense transcription observed in *hrp3Δ* or *hrp1Δhrp3Δ*. Furthermore, a strong inverse correlation ($r = -0.95$) was observed between the decrease in nucleosome occupancy in this region and the antisense PRO-seq peak in each mutant (**Fig 3F**), indicating that disrupted nucleosome organization is strongly associated with increased antisense transcription. These results suggest that, even in WT, cryptic As-TSSs are marked by nucleosomes that are more well-positioned compared to the surrounding nucleosomes. However, in *hrp3Δ* or *hrp1Δhrp3Δ* mutants, the nucleosome-depleted region immediately upstream is far more exaggerated and pronounced, likely contributing to the increased exposure of these cryptic promoter regions and the elevated levels of antisense transcription. Additionally, these findings suggest that while Hrp1 plays a minor role in maintaining nucleosome positioning, Hrp3 is the major remodeler responsible for actively positioning nucleosomes to block cryptic promoters and suppress spurious antisense initiation.

The *hrp1Δhrp3Δ* mutant enabled us to examine how nucleosome organization changes when Hrp3-dependent phasing is lost. Previous studies have identified genomic DNA composition as a major predictor of nucleosome occupancy and binding site preference [75–77]. Consistent with a previous report [78], we observed that nucleosomes in WT *S. pombe* align with regions of higher AT content (**S4A Fig**). However, in *hrp1Δhrp3Δ* mutants, nucleosome dyad positioning became less enriched over AT-rich regions, with an increase in the mean GC content of the associated DNA from 37.8% to 42.8% (S4B in **S4 Fig**). When aligned at the As-TSS, the loss of nucleosome occupancy in the upstream region in mutants is also associated with an AT-rich region (S4C in **S4 Fig**). Together, these results suggest that, in the absence of Hrp3, AT-rich regions within genes become increasingly nucleosome-depleted, thereby creating nucleosome-free regions (NFRs) that we expect to contain promoters facilitating antisense transcription initiation.

To assess whether these regions exhibit promoter-like features, we analyzed their sequence composition and motif content. The mean GC content of antisense promoters was 33.5%, comparable to sense promoters of protein-coding genes (PCGs) in fission yeast (32.3%) and distinct from gene-body regions (37.8%; S5A in **S5 Fig**). We next quantified the fraction of promoters containing at least one TATA-box (consensus TATAWAWR; S5B in **S5 Fig**). 19.2% of antisense

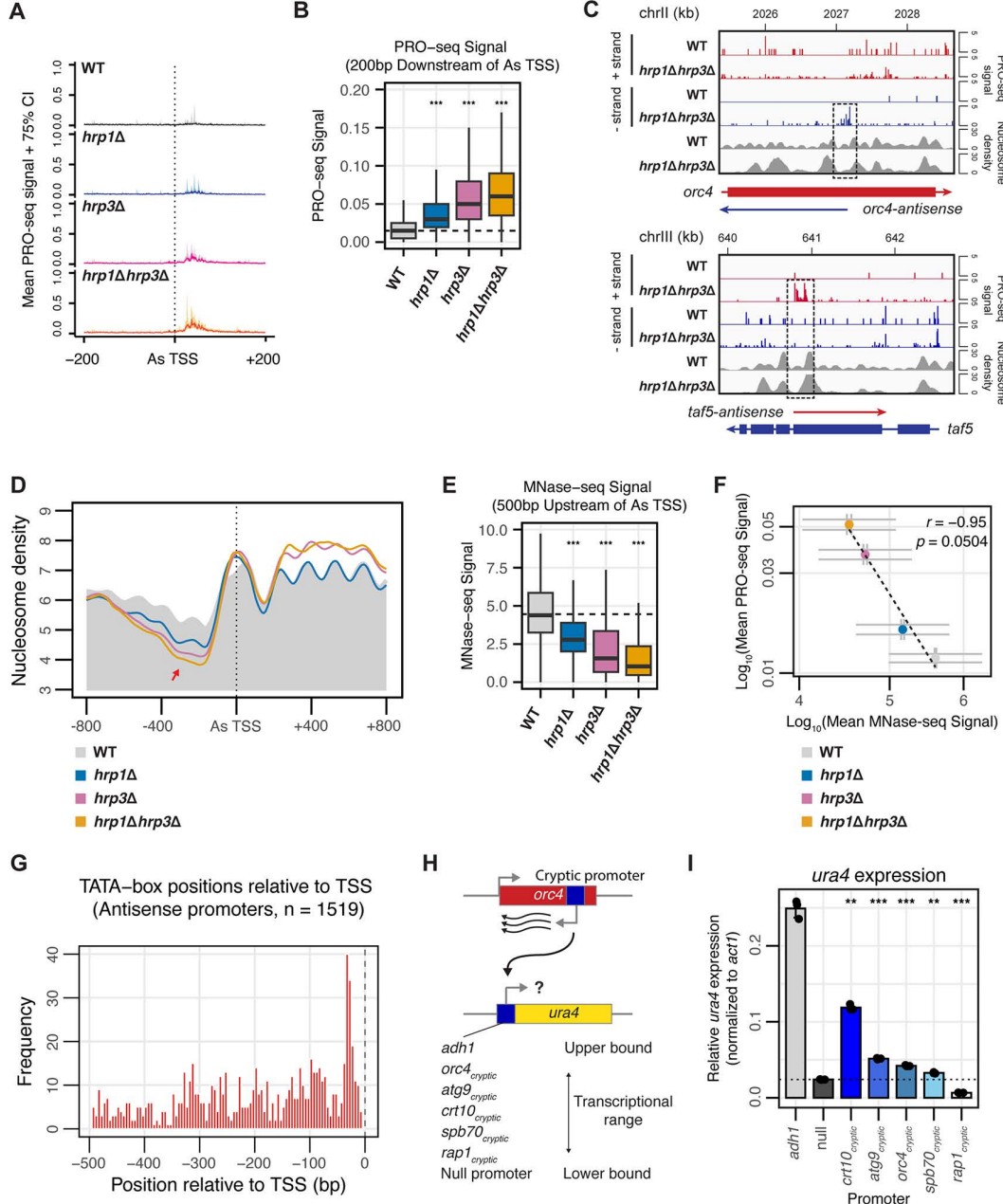

**Fig 3. Hrp3 Blocks Cryptic Promoters to Suppress Antisense Transcription. (A)** Metagene representation of mean PRO-seq signals (solid lines) with 75% confidence intervals (shaded regions) for WT, *hrp1Δ*, *hrp3Δ*, and *hrp1Δhrp3Δ* mutants across the antisense (As) TSS of 4,678 differentially expressed antisense transcripts. Profiles include 200 bp upstream and downstream of the As-TSS. Data represent the mean signal from two biological replicates. **(B)** Boxplot of the mean PRO-seq signal in the 200 bp region downstream of the As-TSS of 4,678 differentially expressed antisense transcripts in WT, *hrp1Δ*, *hrp3Δ*, and *hrp1Δhrp3Δ*. The dotted line represents the median PRO-seq signal in WT. Statistical analysis was performed using ANOVA with Tukey's Honestly Significant Difference (HSD) test. Asterisks indicate statistical significance: $p < 0.05$ (*), $p < 0.01$ (**), $p < 0.001$ (***), n.s. (not significant). **(C)** Genome browser track of PRO-seq and MNase-seq data for *orc4* (top) and *taf5* (bottom), representatives of genes with increased antisense transcription in *hrp1Δhrp3Δ* mutants. PRO-seq signals for the + strand (red) and – strand (blue) are displayed. In the gene schematic, solid boxes represent exons, whereas solid lines represent the entire transcript. The dotted box highlights the region where antisense transcription initiates in the *hrp1Δhrp3Δ* mutant, coinciding with disrupted nucleosome positioning. **(D)** Metaplots of normalized nucleosome density from MNase-seq centered on the As-TSS of 4,678 differentially expressed antisense transcripts in WT, *hrp1Δ*, *hrp3Δ*, and *hrp1Δhrp3Δ*. The WT profile is shown in solid grey. The red arrow indicates a loss of nucleosome occupancy preceding the As-TSS. Profiles include 800 bp upstream and downstream of the As-TSS. **(E)** Boxplot of the mean MNase-seq signal in the 500 bp region upstream of the As-TSS of 4,678 differentially expressed antisense transcripts in WT,

*hrp1Δ*, *hrp3Δ*, and *hrp1Δhrp3Δ*. The dotted line represents the median MNase-seq signal in WT. Statistical analysis was performed using ANOVA with Tukey's Honestly Significant Difference (HSD) test. Asterisks indicate statistical significance: $p < 0.05$ (*), $p < 0.01$ (**), $p < 0.001$ (***), n.s. (not significant). **(F)** Scatterplot of mean PRO-seq signal in the 200 bp region downstream of the antisense TSS versus mean MNase-seq signal in the 500 bp region upstream of the antisense TSS for WT, *hrp1Δ*, *hrp3Δ*, and *hrp1Δhrp3Δ*. Data is plotted for all 4,678 differentially expressed antisense transcripts between WT and *hrp1Δhrp3Δ*. Spearman's correlation and linear regression analysis were performed. Error bars represent the standard error of the mean (SEM). **(G)** Genomic positions of TATA-box hits relative to the antisense TSS for antisense promoters. Histogram shows the distribution of TATA position weight matrix (PWM) match centers (5 bp bins) within −500 to −1 bp upstream of the TSS (FIMO, $p \le$ 1e-4). The dashed vertical line marks the TSS (0 bp). **(H)** Schematic of the *ura4* + reporter system, where the *ura4* promoter is replaced with antisense transcription promoters from five different genes, alongside the normal *adh1* promoter (positive control) and no promoter (negative control). Constructs were tested for their ability to drive *ura4* expression via RT-qPCR. **(I)** Bar plot showing expression of the *ura4* gene relative to the *act1* gene by RT-qPCR ($2^{-\Delta Cq}$) under five different antisense promoters (from **H**). The *adh1* promoter serves as a positive control, while a promoter-deletion strain (null) serves as a negative control. Bars represent mean values from three biological replicates; points indicate individual replicates. The dotted line denotes the mean expression level of the promoter-deletion negative control. Statistical significance was determined using Welch's t-tests comparing each promoter construct to the promoter-deletion negative control, with Holm correction for multiple testing. Asterisks indicate statistical significance: $p < 0.05$ (*), $p < 0.01$ (**), $p < 0.001$ (***), n.s. (not significant).

promoters contained at least one TATA-box, comparable to 22.6% among sense PCG promoters (S5C in S5 Fig). The TATA-box positions were also enriched within ~50 bp upstream of the As-TSS (**Fig 3G**), mirroring the positional distribution observed for sense PCG promoters (S5D in S5 Fig). *De novo* motif discovery within the 500 bp upstream of the As-TSS identified seven significantly enriched sequence motifs (S5E in S5 Fig), each occurring between 178–758 times in the analyzed promoter set (**S1 Table**). Motif-to-TF mapping against a curated *S. pombe* TF motif atlas [79] revealed that six of the seven motifs significantly matched known TF binding specificities, collectively corresponding to 15 of the 38 TFs represented in that atlas (S5F in S5 Fig). Several motifs aligned with more than one TF motif (including Hsf1, Adn3, Sak1, and Fkh2), indicating potential binding redundancy and supporting the capacity of these regions to recruit transcriptional regulators for the initiation of RNAPII transcription. Together, these analyses indicate that the identified antisense promoters share sequence features and core promoter architecture typical of sense promoters in fission yeast.

To test whether the antisense-upstream regions can function as promoters, we replaced the endogenous *ura4* promoter with 500-bp regions upstream of the antisense TSSs from five loci (**Fig 3H**) and quantified *ura4* mRNA by RT-qPCR (normalized to *act1*), including a promoter-deleted strain as a negative control and the *adh1* promoter as a positive control. Apart from *rap1*, four of the five regions (*crt10*, *atg9*, *orc4*, *spb70*) produced significantly higher *ura4* mRNA than the negative control, supporting their intrinsic promoter activity (**Fig 3I**).

These findings suggest that Hrp3 positions nucleosomes over genomic regions prone to nucleosome loss, thereby preventing antisense transcription initiation at permissive sites within genes. This conclusion is further supported by three key observations. First, 87% (4,053/4,678) of antisense transcripts initiate within protein-coding genes, while the remaining 13% initiate in nearby intergenic regions and extend into gene bodies, indicating that antisense initiation in *hrp3Δ* requires DNA sequences or nucleosome arrangements specific to gene bodies. Second, these antisense transcripts are distributed homogeneously across all chromosomes (S4D in **S4 Fig**), suggesting that regions permissive to antisense initiation are cryptically embedded throughout most genes in the genome. Finally, antisense transcripts are also more likely to originate within longer genes (**S4E Fig**). Taken together, these findings show that Hrp3 is essential for maintaining gene body nucleosome phasing and suppressing pervasive cryptic antisense transcription.

### Hrp3-mediated antisense repression links nested gene architecture to meiotic gene expression in S. *pombe*

Given that many antisense transcripts were driven from promoter-like sequences and natural antisense RNAs can have regulatory functions across species [10,12,80], we next asked whether Hrp3-regulated antisense transcripts might have been co-opted for specific gene regulatory programs. We performed K-means clustering of sense genes upregulated in *hrp3Δ* mutants compared to WT and mutants of other remodelers (S6A in S6 Fig). This revealed a distinct group (cluster 4) enriched for genes involved in sexual reproduction (GO:0019953), whereas other clusters were depleted for these genes (S6B in S6 Fig). Concordantly, *hrp3Δ*-upregulated genes significantly overlapped a published list of meiotically

upregulated genes (MUGs) [81] (**Fig 4A** and **4B**), including loci central to sporulation, septation, and meiotic transcription (e.g., *spo6*, *spn6*, *atf21*; **S2 Table**). A notable architectural feature of the MUGs affected in *hrp3Δ* was their frequent occurrence as convergently nested genes—protein-coding genes entirely embedded within, and transcribed opposite to, longer host genes (e.g., *spo6* in *lam2*, *mfr1* in *tfg2*, *meu27* in *chk1*; **Figs 4C**, S6C, and S6D in **S6 Fig**). Genome-wide, we catalogued 212 such convergently nested genes in *S. pombe.* Hypergeometric tests showed significant pairwise overlaps among three sets of genes (MUGs, convergently nested genes, and *hrp3Δ*-upregulated genes), including a significant three-way intersection (**Fig 4D**). Volcano plots further showed that *hrp3Δ*-upregulated genes includes both nested genes and, specifically, nested MUGs (**Fig 4E**). Consistent with a regulated gene expression program, *hrp3* expression was lowest five hours after meiotic induction, coincident with peak MUG expression (**Fig 4F**), which suggests that Hrp3 normally limits the expression of nested meiotic genes during vegetative growth.

To assess whether the loss of CHD remodelers has functional consequences during meiosis, we examined meiotic progression phenotypes in *hrp3Δ*, *hrp1Δ*, and *hrp1Δhrp3Δ* mutants. Upon mating and sporulation, all mutants displayed reduced tetrad formation and increased accumulation of conjugated and non-participant cells relative to WT (S6E and S6F in **S6 Fig**). As a proxy for meiotic efficiency, we calculated the ratio of tetrads to conjugated cells and observed impaired meiotic progression in all mutants (S6G in **S6 Fig**). Notably, *hrp1Δ* exhibited pronounced meiotic defects despite comparatively lower levels of antisense transcription, suggesting that meiotic progression does not follow a simple quantitative relationship with global antisense deregulation. These observations indicate that while Hrp3-dependent antisense repression contributes to the regulation of convergently nested meiotic genes during vegetative growth, meiotic progression likely requires additional molecular pathways or paralog-specific properties of CHD remodelers. Thus, CHD remodeler activity represents a context-dependent regulatory layer within a broader chromatin and transcriptional framework coordinating nested and meiotic gene programs in fission yeast.

### The CHCT domain of Hrp3 prevents antisense transcription

To explore the paralog-specific mechanisms underlying Hrp3 function, we next investigated the molecular basis for Hrp3's suppression of antisense transcription. We leveraged the functional contrast with Hrp1, which exhibited a dramatically smaller antisense phenotype (**Fig 1D**). We first compared their protein sequences and found the greatest divergence in their C-terminal regions (S7A in **S7 Fig**). Detailed annotation using a Pfam search identified the CHD1 helical C-terminal (CHCT) domain [82] as a domain of unknown function (DUF4208; residues 1293–1388) in Hrp3, which was absent in Hrp1 (**Fig 5A**). Structural predictions using AlphaFold showed that the CHCT domain of Hrp3 forms a structured bundle of five α-helices (S7B and S7D in **S7 Fig**), consistent with previous NMR spectroscopy studies [82], and is highly conserved among CHD-family proteins in other eukaryotes. In contrast, the C-terminus of Hrp1 had no predicted structure (S7C and S7E in **S7 Fig**) and has diverged from Hrp3 within the fungal *Schizosaccharomyces* lineage [83]. These findings highlight the structural and evolutionary divergence of the C-terminal regions, which likely underlies the functional differences between Hrp1 and Hrp3.

To test whether the CHCT domain is a key driver of Hrp3's impact on antisense transcription, we generated four chimeric mutants by swapping either the chromodomains (CD) or C-termini (CT) between the two paralogs, knocking them into the *hrp1Δhrp3Δ* background (S7F in **S7 Fig**). We then subjected these mutants to stress conditions targeting DNA damage repair (bleomycin [84]), cell cycle/chromatin (caffeine [85]), the anti-fungal agent clotrimazole [86], protein folding (guanidinium hydrochloride [87]), and transcription and replication (mycophenolic acid [88]), assaying their growth relative to WT, *hrp1Δ*, *hrp3Δ*, and *hrp1Δhrp3Δ* reference strains (**Fig 5B**). In this assay, if Hrp3 function is complemented, then chimeras should resemble the *hrp1Δ*, whereas if they do not, then they should resemble the *hrp3Δ*. Chromodomains were chosen as a control because, unlike the C-termini, they are relatively conserved between Hrp1 and Hrp3 (**S7A Fig**). Whereas *hrp3Δ* and *hrp1Δhrp3Δ* cells exhibited significantly reduced growth under all tested stress conditions compared to WT, with particularly pronounced sensitivity to caffeine, bleomycin, and mycophenolic acid, *hrp1Δ* cells exhibited minimal to no

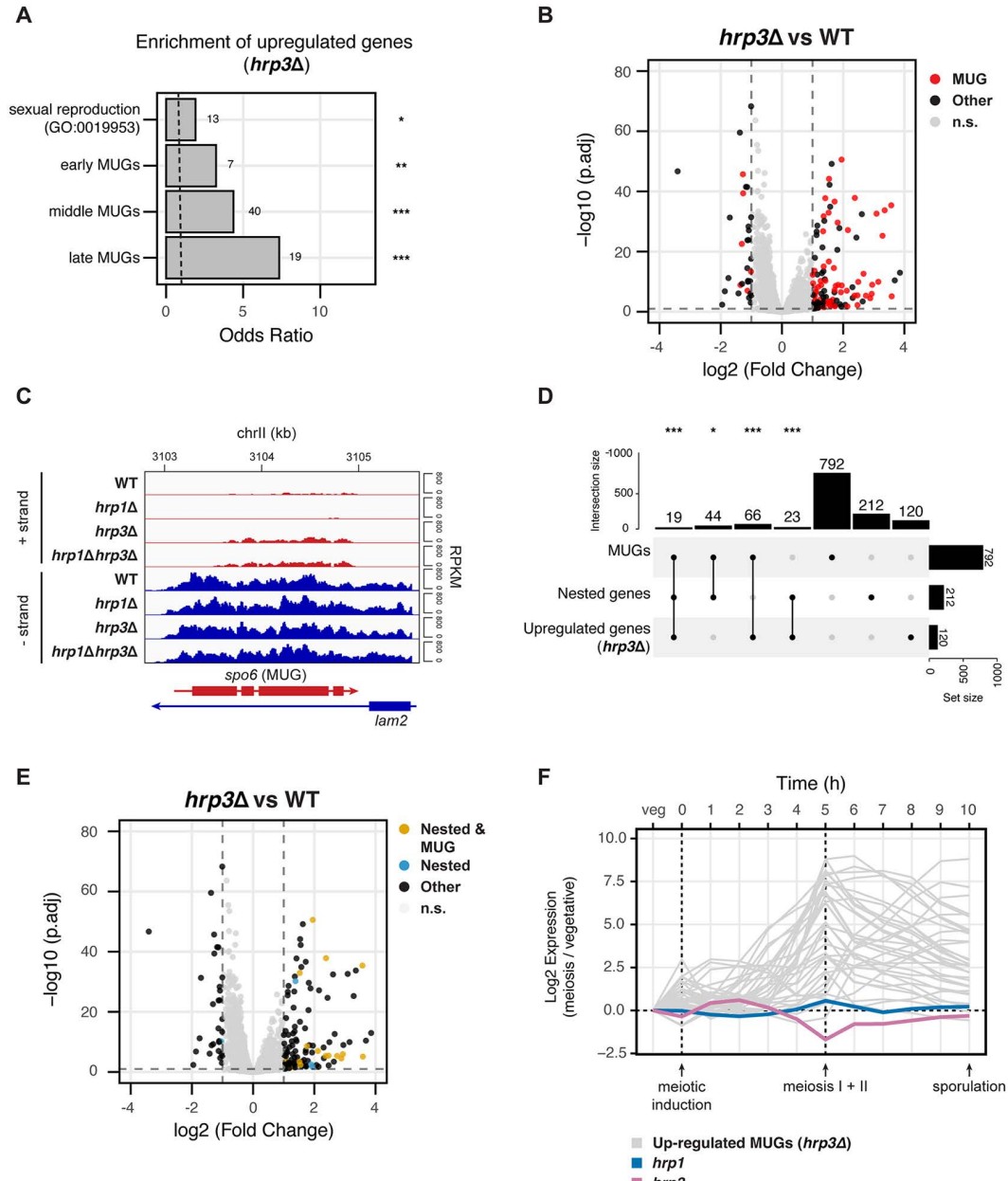

**Fig 4. Hrp3-Mediated Antisense Repression Links Nested Gene Architecture to Meiotic Gene Expression in *S. pombe*. (A)** Enrichment analysis of upregulated genes in the *hrp3Δ* mutant for the sexual reproduction GO term (GO:0019953) and early, middle, and late meiotically upregulated genes (MUGs) as reported by Mata et al. The dashed line indicates an odds ratio of 1. Statistical analysis was performed using Fisher's exact test. Asterisks indicate statistical significance: p<0.05 (*), p<0.01 (**), p<0.001 (***), n.s. (not significant). **(B)** Volcano plot representing the differentially expressed genes in *hrp3Δ* versus WT. MUGs are highlighted in red, while all other genes are in black. Genes in grey do not meet the thresholds for both significance (-log$_{10}$(p.adj) > 1) and effect size (-1<log$_2$FC<1). **(C)** Genome browser track of mRNA-seq data for the *spo6* and *lam2* genes in WT, *hrp1Δ*, *hrp3Δ* and *hrp1Δhrp3Δ*. Tracks are separated into the + strand (red) and – strand (blue), with values shown in RPKM. The MUG within the pair is denoted in brackets. **(D)** UpSet plot showing the overlap between MUGs, convergent nested genes and upregulated genes in *hrp3Δ*. Statistical analysis was performed using the hypergeometric test. Asterisks indicate statistical significance: p<0.05 (*), p<0.01 (**), p<0.001 (***), n.s. (not significant). **(E)** Volcano plot representing the differentially expressed genes in *hrp3Δ* versus WT. Genes which are both convergently nested and MUGs (n=19) are highlighted in gold, other convergent nested genes (n=193) are highlighted in blue, and all other genes are in black. Genes in grey do not meet the thresholds for both significance (-log$_{10}$(p.adj) > 1) and effect size (-1<log$_2$FC<1). **(F)** Line plot showing the temporal expression of *hrp1*, *hrp3*, and 35 MUGs upregulated in *hrp3Δ* at 1-hour intervals during meiosis in *S. pombe*. Data were obtained from Mata et al. [81].

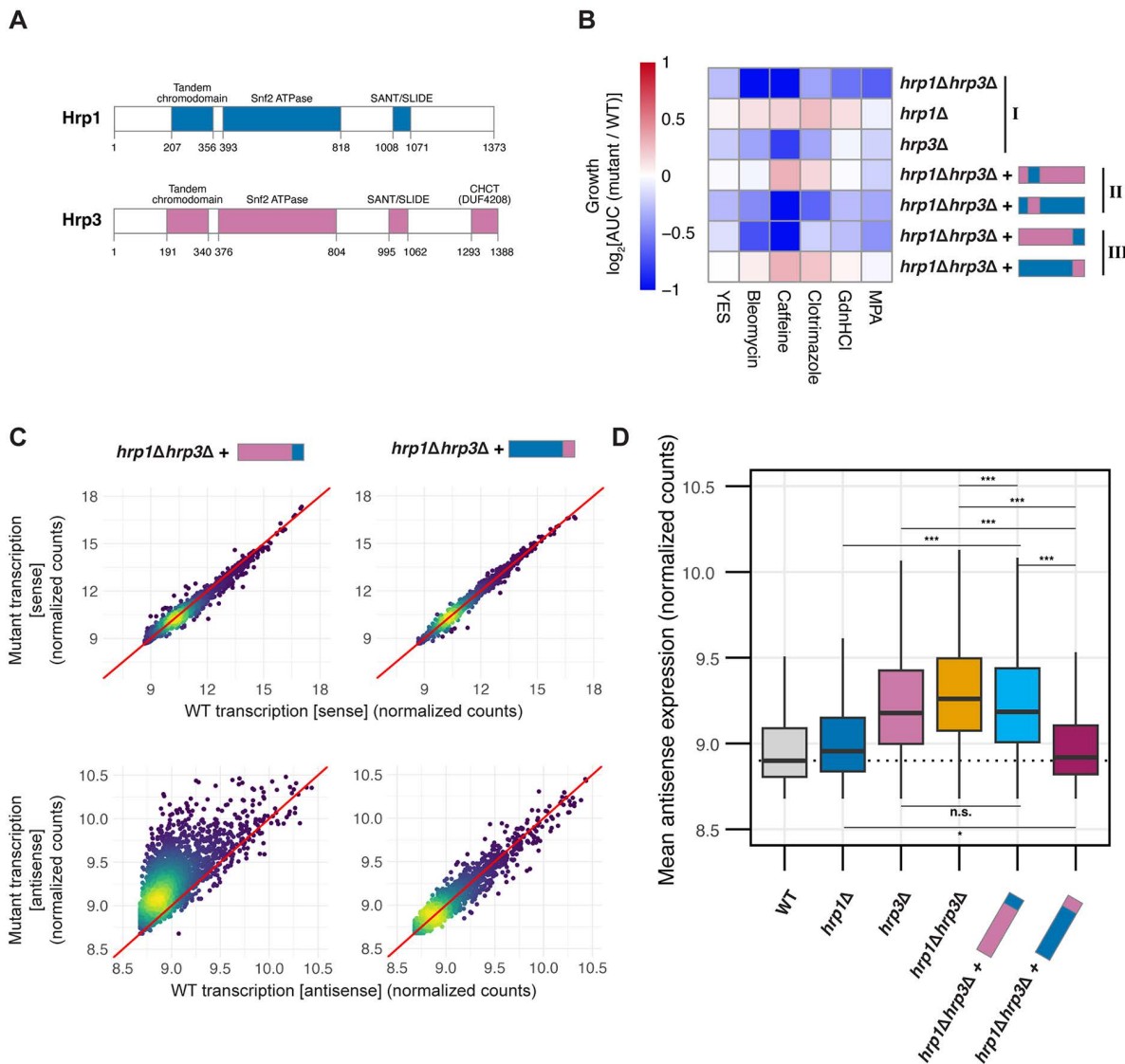

**Fig 5. The CHCT Domain of Hrp3 Prevents Antisense Transcription. (A)** Heatmap of growth phenotypes across four biological replicates when grown in YES medium alone or supplemented with 25 ng/µl bleomycin, 14 mM caffeine, 200 ng/ml clotrimazole, 5 mM guanidine hydrochloride (GHCl), or 25 µg/ml mycophenolic acid (MPA), represented by $\log_2$AUC values of mutants relative to WT. Roman numerals indicate mutant groups: (I) deletion mutants (II) chromodomain-swapped chimeras in the *hrp1Δhrp3Δ* mutant background, and **(III)** C-terminus-swapped chimeras in the *hrp1Δhrp3Δ* mutant background. Data represent the mean growth from four biological replicates. **(B)** Schematic representation of the protein structures of Hrp1 and Hrp3, including domain annotations with residue start and end positions. **(C)** Density-colored scatterplots representing mutant (*hrp1Δhrp3Δ::hrp3-CT_hrp1* or *hrp1Δhrp3Δ::hrp1-CT_hrp3*) transcription against WT transcription for either sense transcripts (top row) or antisense transcripts (bottom row) for all protein-coding genes. Count data is derived from variance-stabilizing transformation (vst) of raw mRNA-seq counts from three biological replicates. **(D)** Boxplot of normalized antisense expression counts (as calculated in C) for WT, *hrp1Δ*, *hrp3Δ*, *hrp1Δhrp3Δ*, *hrp1Δhrp3Δ::hrp3-CT_hrp1* and *hrp1Δhrp3Δ::hrp1-CT_hrp3*. The dotted line represents the median antisense expression in WT. Statistical analysis was performed using ANOVA with Tukey's Honestly Significant Difference (HSD) test. Asterisks indicate statistical significance: $p < 0.05$ (*), $p < 0.01$ (**), $p < 0.001$ (***), n.s. (not significant).

growth defects relative to WT (**Fig 5B**; **Group I**). This pattern mirrors the extent of nucleosome disruption and antisense transcription in these mutants, suggesting that elevated antisense transcription imposes a broad fitness cost. Swapping the chromodomains between Hrp1 and Hrp3 did not alter the phenotype, indicating functional interchangeability (**Fig 5B**;

**Group II**). By contrast, replacing the Hrp3 C-terminus with that of Hrp1 (*hrp1Δhrp3Δ::hrp3-CT$_{hrp1}$*) recapitulated the growth defects of *hrp3Δ* (**Fig 5B**; **Group III**), while introducing the Hrp3 CHCT domain onto Hrp1 (*hrp1Δ hrp3Δ::hrp1-CT$_{hrp3}$*) was sufficient to rescue the growth phenotype, even in the absence of Hrp3 (**Fig 5B**; **Group III**), suggesting that the CHCT domain plays a critical role in Hrp3 function.

To determine whether the CHCT domain's phenotypic impact was rooted in regulating antisense transcription, we next performed mRNA-seq on the *hrp1Δhrp3Δ::hrp3-CT$_{hrp1}$* and *hrp1Δhrp3Δ::hrp1-CT$_{hrp3}$* mutants (**Fig 5C and 5D**). Consistent with the growth phenotypes, transplanting the C-terminus of Hrp1 onto Hrp3 (*hrp1Δhrp3Δ::hrp3-CT$_{hrp1}$*) resulted in elevated levels of antisense transcription, closely resembling the *hrp3Δ* mutant. By contrast, swapping the CHCT domain of Hrp3 onto Hrp1 (*hrp1Δhrp3Δ::hrp1-CT$_{hrp3}$*) did not lead to such an increase. These findings indicate that the CHCT domain of Hrp3 is essential for the regulation of antisense transcription in cells.

### The CHCT domain of Hrp3 interacts with the transcriptional regulator Prf1

Although not fully resolved in cryo-electron microscopy studies [89,90], the terminal location of the CHCT domain of Chd1, distal from the nucleosome, suggested it may mediate protein-protein interactions. To investigate this possibility, we performed an *in silico* interaction screen using Alphafold2 Multimer [91] on the CHCT domain with a set of 2,692 nuclear genes in *S. pombe* [92]. Our results identified 12 candidates with an average interface predicted template modelling (IPTM) score of > 0.5 (S8A in **S8 Fig and S3 Table**). We filtered the predicted interactors to include only proteins associated with transcription elongation, as CHD1 is primarily involved in this process. Among these, the transcriptional regulator Prf1, the *S. pombe* homolog of the conserved RTF1 protein found in other eukaryotes, emerged as the strongest candidate (ipTM = 0.5516). The same interaction was predicted between full-length Prf1 and Hrp3 (S8B and S8C in **S8 Fig**). Notably, while Prf1/Rtf1 is a stable component of the elongation-associated Paf1 complex in budding yeast, it is biochemically distinct in metazoans and in *S. pombe* [93]. Nevertheless, our results are consistent with previous reports identifying the Paf1 complex as a potential Chd1 interactor in budding yeast [94,95]. In contrast, a similar prediction for Prf1 and Hrp1 did not reproduce the interaction (S8D in **S8 Fig**). The predicted Hrp3–Prf1 interface consisted of a distal binding region and a hydrophobic pocket within the CHCT domain, comprising three contact points (R2, R3, R4) and a distal region (R1) that surround a small hydrophobic α-helix at the N-terminus of Prf1 (N-helix) (**Fig 6A and 6B**). These contact points are conserved across major eukaryotic lineages (S9A and S9B in **S9 Fig**). Further AlphaFold predictions with CHD1 from *Homo sapiens* and *Arabidopsis thaliana* also supported an interaction between the CHCT domain and the Prf1/RTF1 N-terminal helix (S8E–S8H in **S8 Fig**), suggesting possible evolutionary conservation of this mechanism.

To validate the predicted interaction in cells, we first performed endogenous co-immunoprecipitation from strains expressing Prf1-FLAG and Hrp3-Myc. Anti-FLAG IP recovered a modest but reproducible amount of Hrp3-Myc (S8I in **S8 Fig**), consistent with a specific but likely transient association *in vivo*. We next asked whether the interaction is sufficient outside its native context. In bacterial co-expression assays, GST-tagged Hrp3-CHCT pulled down His-tagged full-length Prf1/RTF1 (**Fig 6D and 6E**), and this extended to orthologs from both *Homo sapiens* and *Arabidopsis thaliana* (S8J in **S8 Fig**), supporting a conserved interaction. Finally, to experimentally define the interface with precision, we repeated the interaction with mutants guided by AlphaFold. Compared to the interaction between wild-type GST–CHCT and His–Prf1, alanine substitutions in the Prf1 N-terminal helix (but not in a distal region) abolished binding (**Fig 6D**). Conversely, alanine substitutions in the CHCT interface regions (R1–R4) disrupted binding to wild-type Prf1 (**Fig 6E**). Additionally, when we extended these findings to test the CHCT mutants' functional impact, strains expressing Hrp3-CHCT domain mutants displayed stress sensitivity comparable to *hrp3Δ* and CHCT-deleted strains (**Fig 6F**). Together with a recent study [96] independently reporting a similar interaction with mouse CHD1/2 and RTF1, these results demonstrate that the Hrp3/CHD1 CHCT domain mediates a conserved interaction with Prf1/RTF1 and that this interface is critical for cellular fitness under stress.

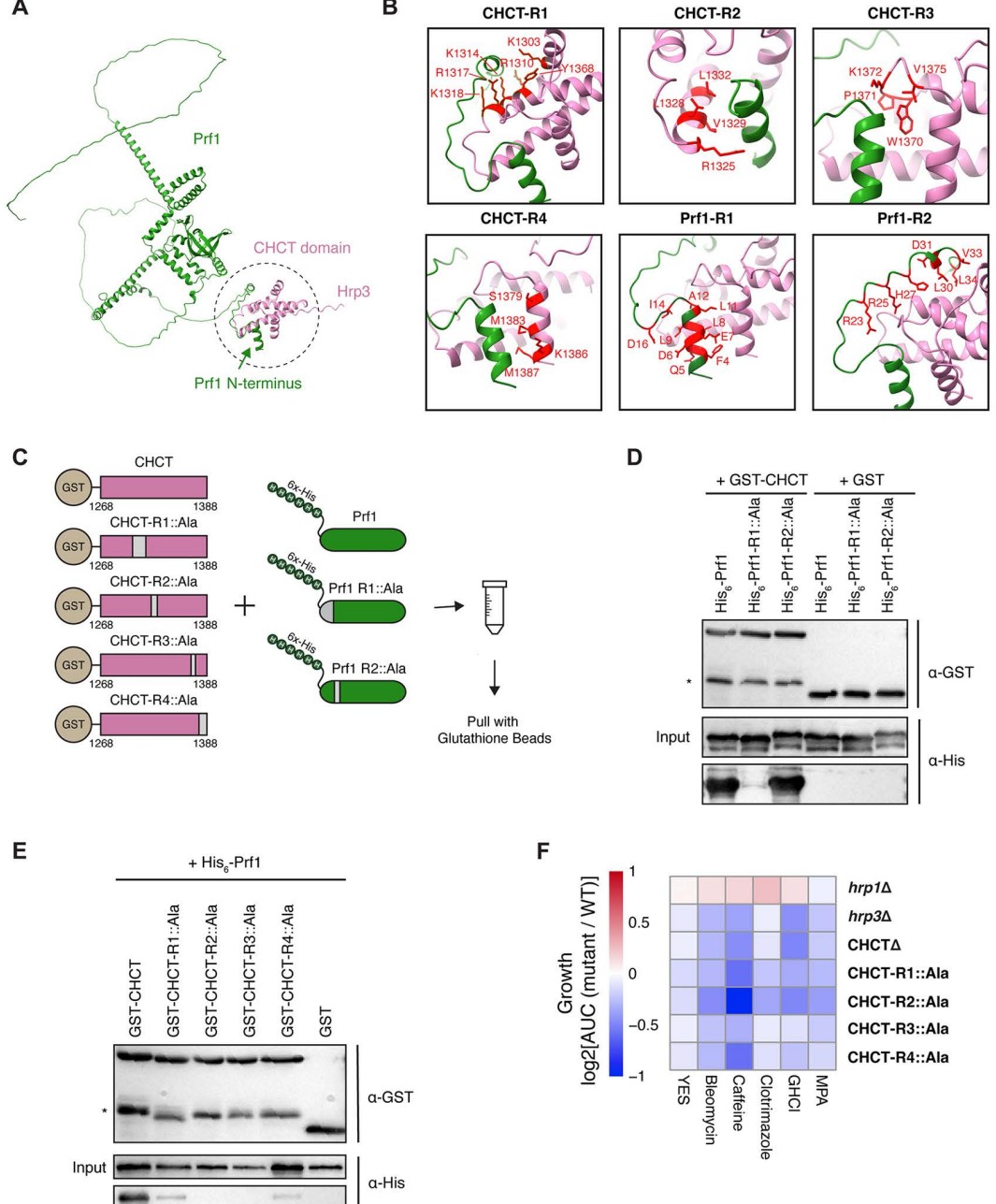

**Fig 6. The CHCT Domain of Hrp3 Interacts with the Transcriptional Regulator Prf1. (A)** AlphaFold2-predicted structure of the interaction between the N-terminus of Prf1 and the CHCT domain of Hrp3. The predicted binding interface is highlighted with a dotted circle. **(B)** Close-up views across six AlphaFold2-predicted binding regions between Prf1 (green) and the CHCT domain (pink). Residues involved in the interaction are highlighted in red for each region. **(C)** Schematic of the in vitro interaction assay. Binding was tested between GST-tagged CHCT (wild-type and alanine mutants) and His-tagged Prf1 (wild-type and alanine mutants). Regions of the proteins with residues mutated to alanine are shown in grey. **(D)** Western blot showing anti-GST and anti-His signals for GST-pulldowns of GST-CHCT (wild-type or mutated) and wild-type His-Prf1. Asterisk (*) denotes truncated GST-CHCT constructs. **(E)** Western blot showing anti-GST and anti-His signals for GST-pulldowns for wild- type GST-CHCT with His-Prf1 (wild-type or mutated). Asterisk (*) denotes truncated GST-CHCT constructs. **(F)** Heatmap of growth phenotypes of *hrp1Δ, hrp3Δ, hrp3-CHCTΔ*, and *in vivo* alanine-swapped CHCT mutants (as in panel C) when grown in YES medium alone or supplemented with 25 ng/µl bleomycin, 14 mM caffeine, 200 ng/ml clotrimazole, 5 mM guanidine hydrochloride (GHCl), or 25 µg/ml mycophenolic acid (MPA). Data represent the $\log_2$AUC values of mutant growth relative to WT under various stress conditions across four biological replicates.

## Prf1-dependent recruitment of Hrp3 establishes H2Bub and nucleosome phasing to suppress antisense transcription

We next investigated the impact of the Hrp3-Prf1 interaction on chromatin and transcriptional regulation. Prf1/RTF1 is enriched across gene bodies and is essential for recruitment of the H2B ubiquitination machinery via its histone modifying domain (HMD) [97], a crucial step in the activation of the histone chaperone FACT. We hypothesized that the Hrp3–Prf1 interaction synergizes to regulate H2Bub deposition specifically during transcription elongation, ensuring retention of nucleosomes in the wake of transcription. To test this, we first performed ChIP-seq to examine the genomic distribution of both Hrp3, Hrp1, and H2BK119ub, using the known elongation mark H3K36me3 as a reference. In contrast to Hrp1, Hrp3 localized across gene bodies with enrichment profiles similar to the profiles of the elongation marks H3K36me3 and H2Bub (**Fig 7A**), and this localization was correlated with gene expression (**Fig 7B**). The contrast between Hrp1 and Hrp3's genomic profiles suggested that the CHCT domain of Hrp3 may drive its co-localization to zones of active elongation. To test this, we performed Hrp3 ChIP-seq in a *prf1Δ* background, finding a marked reduction in gene body enrichment (**Figs 7C–7D and S10C**, Fisher's exact test, $p < 2.2e-16$), and establishing that Prf1 is a key determinant of Hrp3's genomic localization.

We next investigated the functional link between Hrp3, Prf1, and nucleosome positioning. We first profiled H2BK119ub in WT, *hrp1Δ*, *hrp3Δ*, *hrp1Δhrp3Δ*, and *prf1Δ* (**Fig 7E**). As expected, *prf1Δ* showed a pronounced loss of H2Bub across gene bodies. By contrast, *hrp3Δ* and *hrp1Δhrp3Δ* also displayed significant, albeit less severe, reductions, whereas *hrp1Δ* alone did not (S10D in S10 Fig). These results support a hierarchy in which Prf1 functions upstream of Hrp3, with the Hrp3 CHCT domain facilitating Prf1-dependent H2Bub deposition. The effect was modestly stronger at highly expressed genes (S10E in S10 Fig), consistent with a tight coupling of Prf1/Hrp3 activity to transcription elongation.

As H2Bub is known to enhance nucleosome stability [98], we next assessed the impact of the loss of *prf1* on nucleosome organization using MNase-seq. Nucleosome phasing was severely disrupted in the *prf1Δ* mutant, as well as in the *hrp1Δhrp3Δprf1Δ* triple mutant (**Fig 7F**). Given the similar degree of nucleosome disruption observed in both mutants, this provides further support that Prf1 and Hrp3 act in the same pathway to maintain gene-body nucleosome spacing. Consistent with this, both the *prf1Δ* and *hrp1Δhrp3Δprf1Δ* mutants exhibited increased antisense transcription, comparable to the *hrp1Δhrp3Δ* mutant (**Fig 7G**), extending the epistasis to the regulation of cryptic antisense initiation (**Fig 7H**). Similar to what was observed for *hrp3Δ*, sense transcription was largely unchanged in *prf1Δ* and *hrp1Δhrp3Δprf1Δ* (S10F and S10G in S10 Fig). Notably, the *prf1Δ* and *hrp1Δhrp3Δprf1Δ* mutants displayed growth defects measured by endpoint cell density compared to the *hrp1Δhrp3Δ* mutant, indicating that Prf1 has additional essential roles beyond its cooperation with Hrp3 in cells [99] (S10H in S10 Fig). Together, these findings support a surveillance model in which Prf1 recruits Hrp3 to gene bodies during active elongation, and that Prf1 and Hrp3 then cooperate to establish nucleosome organization critical for suppressing cryptic antisense transcripts.

## Discussion

In this study, we demonstrate that the CHD-family nucleosome remodeler Hrp3 is a central regulator of cryptic antisense transcription. Hrp3 establishes nucleosome arrays within gene bodies, suppressing cryptic antisense initiation by masking embedded promoter-like regions within genes. While many antisense transcripts likely reflect opportunistic expression of otherwise neutral sequence variation, a defined subset corresponds to convergently nested meiotic genes whose expression during vegetative growth is associated with Hrp3-dependent antisense repression. Mechanistically, the Hrp3 CHCT domain engages the elongation factor Prf1/RTF1, and loss of Prf1 redistributes Hrp3 away from gene bodies, effectively enabling the elongation machinery to recruit the CHD-family nucleosome remodelers onto chromatin. This coupling provides spatial and temporal control of nucleosome organization during transcription and safeguards promoter fidelity. Our results build on earlier observations [65–68] that *hrp3Δ* disrupts gene-body phasing and elevates antisense transcription in *S. pombe* by defining where cryptic antisense transcripts initiate, identifying a Prf1-dependent recruitment pathway, and

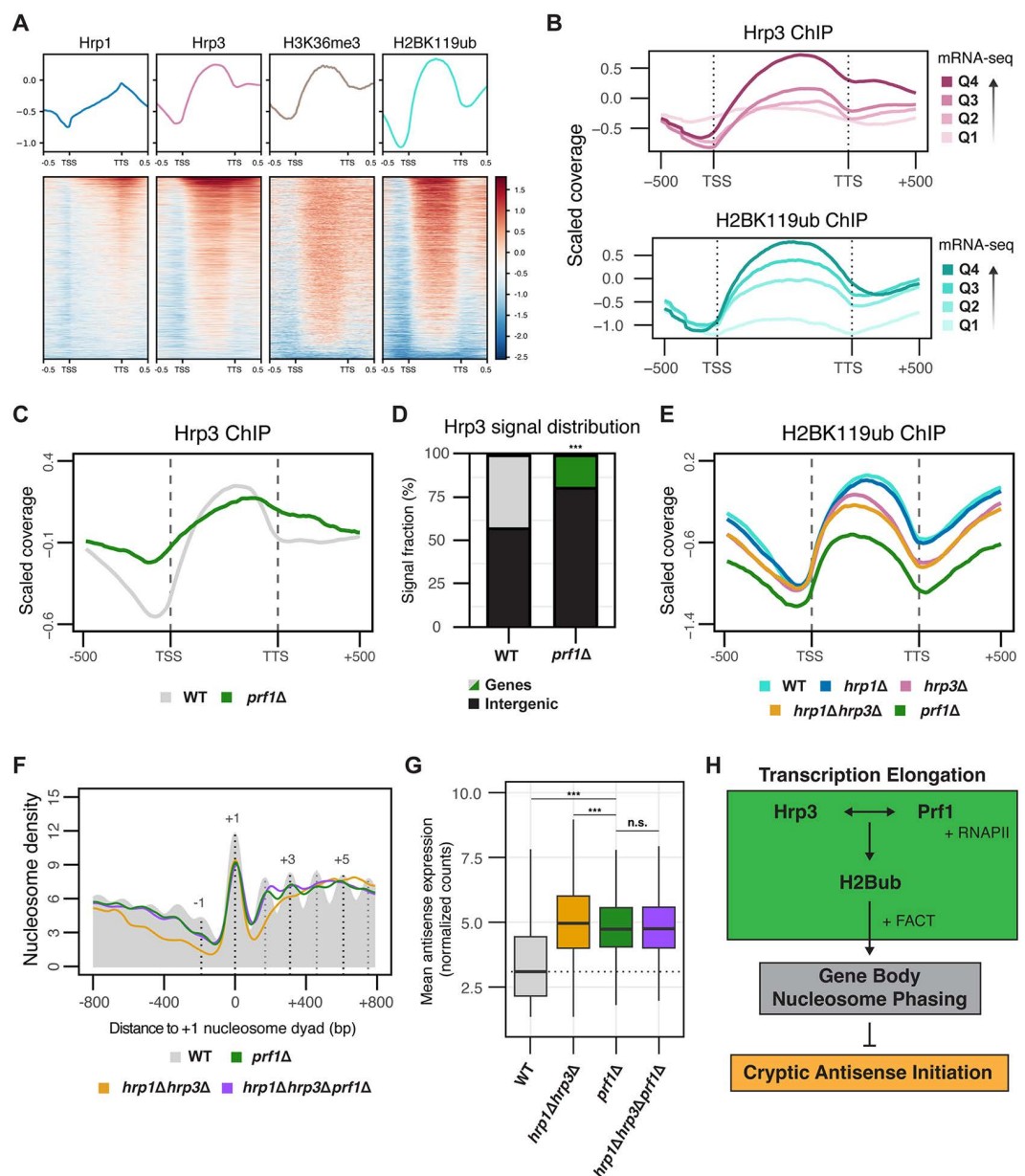

**Fig 7. Prf1-dependent Recruitment of Hrp3 Establishes H2Bub and Nucleosome Phasing to Suppress Antisense Transcription. (A)** Metagene profiles and heatmaps showing ChIP-seq enrichment over input for Hrp1, Hrp3, H3K36me3 and H2BK119ub across all protein-coding genes. For Hrp1 and Hrp3, data were obtained using an anti-myc antibody in WT strains carrying myc-tagged Hrp1 and Hrp3. **(B)** Metagene plot of the ChIP-seq coverage relative to input for Hrp3-myc and H2BK119ub in WT across all protein coding genes. Genes were divided into four quartiles based on expression in WT. Quartiles are numbered by expression level, with Q1 being the lowest and Q4 being the highest. **(C)** Metagene profiles showing Hrp3 ChIP-seq enrichment over input for WT and *prf1Δ* across all protein-coding genes. For both strains, data were obtained using an anti-myc antibody in strains carrying myc-tagged Hrp3. **(D)** Stacked bar plot showing genome-wide partitioning of Hrp3 ChIP signal between annotated genes and intergenic regions in WT and *prf1Δ*. Bars show the fraction of total mapped ChIP signal assigned to each category. Statistical analysis was performed using Fisher's exact test on ChIP-seq counts. Asterisks indicate statistical significance: p < 0.05 (*), p < 0.01 (**), p < 0.001 (***), n.s. (not significant). **(E)** Metagene plot of H2BK119ub ChIP-seq coverage relative to input for WT, *hrp1Δ, hrp3Δ, hrp1Δhrp3Δ* and *prf1Δ* across all protein coding genes. **(F)** Metaplots of normalized nucleosome density from MNase-seq centered on the +1 nucleosome dyad for WT, *prf1Δ, hrp1Δhrp3Δ* and *hrp1Δhrp3Δprf1Δ*. The WT nucleosome profile is shown in solid grey, with dotted lines indicating the positions of select nucleosome maxima (-1, +1, +3 and +5). Profiles include 800 bp upstream and downstream of the +1 nucleosome dyad for all protein-coding genes. The plotted data represents the average signal from two biological replicates. **(G)** Boxplot of normalized antisense expression counts for all antisense genes (calculated using vst-normalized raw mRNA-seq counts) for WT, *hrp1Δhrp3Δ, prf1Δ* and *hrp1Δhrp3Δprf1Δ*. The dotted line represents the median antisense expression in WT. Data are calculated from

three biological replicates. Statistical analysis was performed using ANOVA with Tukey's Honestly Significant Difference (HSD) test. Asterisks indicate statistical significance: $p < 0.05$ (*), $p < 0.01$ (**), $p < 0.001$ (***), n.s. (not significant). **(H)** Schematic illustrating the relationship between Hrp3 and Prf1 in the context of H2Bub deposition and nucleosome phasing, and its impact on antisense transcription.

revealing a programmatic link to nested meiotic gene regulation. Together, these findings highlight how coordinated mechanisms spanning elongation and chromatin structure enforce proper promoter licensing and gene expression.

Antisense transcription is an evolutionarily ancient and ubiquitous phenomenon, observed throughout life [5,100], but its evolutionary significance and regulatory mechanisms remain incompletely understood. Furthermore, the evolutionary pressures governing antisense transcription remain a subject of debate [12,101], as its extent appears to be highly species-specific. For instance, 30–40% of genes in humans are bidirectionally transcribed, while this proportion increases to up to 70% in mice [8,102–104]. Here, we expand the catalog of antisense transcripts in *S. pombe*, identifying thousands of dormant antisense RNAs that overlap up to 55% of all protein coding genes. Many of these antisense transcripts, including several meiotic genes, are nested within larger host genes in a convergent orientation and repressed by Hrp3. While most of these antisense transcripts do not appear to directly regulate the sense expression of overlapping genes, as observed in other species [10,12,105,106], they may serve alternative *trans*-regulatory roles or encode functional proteins. This suggests that nature may have co-opted these additional transcripts for diverse functions and regulatory roles in a species-specific manner, akin to how gene duplication and neofunctionalization have driven evolutionary innovation [107–110]. In this way, antisense transcription can be viewed as a source of complexity, providing raw material for the evolution of novel regulatory networks and functional elements. The dual nature of antisense transcription—acting as either beneficial regulatory elements or harmful transcriptional noise—underscores the complexity of transcriptional regulation and the evolutionary pressures that maintain this phenomenon. Unregulated antisense transcription, as seen in *hrp3Δ* mutants, imposes significant fitness costs, particularly under stress conditions, emphasizing the necessity for tight regulatory control.

A key question arising from our study is how Hrp3 achieves the precise regulation of nucleosome positioning in coordination with the transcriptional machinery. Our study, together with recent work in budding yeast [96], identifies the interaction between the Hrp3 CHCT domain and the elongation complex subunit Prf1/RTF1 as a critical link in this process. Loss of Prf1 is accompanied by loss of Hrp3 specificity towards gene bodies, consistent with Prf1 acting upstream to promote H2Bub deposition and to retain Hrp3 during elongation. Disruption of this interaction—either by loss of Hrp3 or Prf1—results in similar phenotypes, including impaired nucleosome phasing and increased antisense transcription. These findings align with previous studies showing that H2Bub, deposited by Prf1/RTF1, facilitates the activity of FACT [98,111], a histone chaperone that reassembles nucleosomes in the wake of RNAPII. Concordantly, FACT loss disrupts spacing and elevates antisense transcription [112–116], supporting a model in which Prf1 (via H2Bub) and Hrp3 cooperatively establish gene-body phasing, with FACT acting as a partner to reinforce nucleosome reassembly. In this framework, Prf1 sits upstream of Hrp3 in the hierarchy: the Prf1 N-terminus engages Hrp3's CHCT to promote its association with elongating gene-body nucleosomes, enabling efficient H2Bub deposition; FACT then works with Hrp3 remodeling to maintain phased arrays that occlude embedded promoter-like sites and suppress cryptic initiation. This coupling helps explain why cryptic antisense TSSs arise predominantly within gene bodies—precisely where elongation-linked H2Bub, FACT activity, and Hrp3 remodeling intersect to enforce promoter licensing.

Although our data support a model in which Prf1/RTF1 and FACT cooperate with Hrp3 to establish gene-body phasing, we do not claim that this is the only route to phasing *in vivo* or that Hrp3 lacks intrinsic spacing capacity. CHD-family remodelers can space nucleosomes *in vitro* [66,117,118], but these studies primarily report sliding-based spacing on pre-assembled arrays. By contrast, our *in vivo* profiles emphasize positional losses and redistributions of specific gene-body nucleosomes, which reflect not only spacing per se but also placement, stabilization, and retention on chromatin. Thus, Hrp3 may be fully capable of autonomous spacing of nucleosomes, while in cells it acts with elongation factors to position

and maintain nucleosomes at critical genomic sites. This framework helps reconcile differences between *in vitro* spacing patterns [118] and *in vivo* MNase-seq profiles and underscores the need to study remodelers in their native context, where interactions with elongation factors and histone chaperones tune when and where remodelers operate.

Additionally, our findings show that chromatin, specifically well-phased nucleosome arrays, play a key role in regulating antisense transcription. Building on previous studies that established the role of nucleosome arrays in suppressing cryptic transcription [33–35], we extend this concept to within gene bodies. We show that cryptic promoters can arise within canonical protein coding genes, which Hrp3 works together with the transcription apparatus to suppress and ensure the fidelity of transcription. The evolution of nucleosome arrays in eukaryotes, alongside the diversification of nucleosome remodeler families as their architects, likely reflect the demand for the increasing complexity of transcriptional regulation during eukaryogenesis [119]. Furthermore, the widespread distribution of AT-rich regions which can function as cryptic antisense promoters within genes appears to be an inevitable consequence of evolutionary constraints, such as the local nucleotide composition imposed by protein-coding sequences. Indeed, studies have shown that up to 36% of transcription factor binding sites in humans lie within genes [120]. Although codon redundancy may have partially evolved to mitigate this issue [121,122], the nucleosome remodeler Hrp3 remains essential for ensuring that these cryptic promoters remain latent unless specifically required. This also suggests that the widespread distribution of cryptic antisense promoters is an inherent feature of eukaryotic genomes, shaped by both sequence composition and chromatin architecture.

In conclusion, our study highlights the critical role of Hrp3 in maintaining proper promoter licensing and antisense transcription regulation in *S. pombe*. By positioning nucleosomes to occlude cryptic antisense promoters and coordinating with the transcription elongation machinery, Hrp3 ensures proper gene regulation and cellular fitness. The thousands of newly identified antisense transcripts revealed in this study likely play context-specific roles that remain to be fully characterized. These findings offer valuable insights into the evolutionary and functional importance of nucleosome remodelers and their pivotal role in shaping the transcriptional landscape of eukaryotic genomes.

## Materials and methods

### Yeast culture, strain manipulation, phenotypic and mating assays

All strains used in this study are listed in **S4 Table**. Yeast (*S. pombe* strain 972) were cultured under standard conditions (32 °C). Unless otherwise stated, media used were either EMM2 (Edinburgh Minimal Media) [123] or YES (Yeast Extract with Supplements) [124].

*S. pombe strains* were manipulated either using one-step PCR-based HR mutagenesis [125] or the SpEDIT CRISPR editing platform [126]. For both, heat-shock was used to transform chemically-competent parent strains [127]. Donor templates for homologous recombination were prepared using either Gibson assembly [128] using reagents supplied by the IMP Molecular Biology Service combined, with site directed mutagenesis, or by direct synthesis from commercial vendors (IDT, Genscript). For commercially synthesized HR templates, sequences were codon optimized according to *S. pombe* codon usage using Benchling (https://benchling.com). In all experiments, WT control transformations were performed such that synonymous mutations were introduced to delete CRISPR gRNA target sites. All manipulations were verified by PCR genotyping and where appropriate sequencing of the relevant locus.

To construct the *ura4*-based reporter, a construct (PJY_135) consisting of homology to the 5′ of the *ura4*-D18 locus [129], the selectable marker natMX6, a ~ 1 kb fragment corresponding to either the S. pombe *adh1* promoter, a ~ 500 bp fragment tested to have minimal transcriptional activity, or ~500 bp fragments corresponding to antisense promoters within *rap1*, *atg9*, *crt10*, *spb70* or *orc4* were amplified from gDNA extracted from WT *S. pombe*, and the *ura4*$^+$ gene were assembled by Gibson. The *ura4-D18* locus was then replaced with the reporter via one-step PCR followed by homologous recombination [125]. Correct integration was verified by PCR genotyping.

Phenotypic assays in this study were conducted using a plate-based bulk growth assay. $OD_{600}$ was continuously monitored in 384-well plates (Nunc) under the stress conditions specified in the figure legends, at the standard growth

temperature with shaking. Saturated cultures were diluted 100-fold for sub-culturing, and all assays were performed in either YES, EMM complete, or EMM without uracil (EMM-Ura, for reporter assays). Data collection was carried out using a BioTek Epoch2 microplate reader. Growth curves were analyzed in R, and the area under the curve (AUC) was calculated using the Growthcurver package (https://cran.r-project.org/web/packages/growthcurver/vignettes/Growthcurver-vignette.html). All subsequent data analysis was done in R.

For mating assays, cells were spotted to Malt Extract (ME) plates and incubated for 24 h at 25°C. For each genotype, 6 random fields of view were imaged from 3 independent replicates at 8 h and 24 h after plating on ME. Images were randomized and scored by two independent researchers blinded to the genotypes. Cells were classified as tetrads, conjugated pairs, or non-participants according to predefined criteria. Any disagreements were resolved by consensus.

## Bacterial protein expression and in vitro pulldown experiments

Sequences corresponding to WT and mutated versions of *prf1* and the *hrp3* CHCT domain (from either *Schizosaccharomyces pombe*, *Arabidopsis thaliana* or *Homo sapiens*) were synthesized directly from commercial vendors (Genscript) and amplified by PCR. The *prf1* sequences were cloned into pET28a (Novagen), while the *hrp3* CHCT domain sequences were cloned into pGEX-4T-1 (Cytiva). Assembly of all plasmids was carried out by Gibson.

The plasmids expressing WT and mutated versions of 6xHis-tagged *prf1* were then co-transformed with plasmids expressing WT or mutated versions of GST-tagged *hrp3* CHCT domain into E. coli BL21 (DE3) RIL. Ten milliliter cultures grown overnight at 37 °C were diluted into 200 ml of fresh LB medium and grown at RT for 3 h and then induced for 5 h at RT. Cells equivalent to 100 of culture were resuspended in 5 ml of TBS containing 0.1% Triton X-100, 1 mM DTT, protease inhibitors (Roche), 10 µl of benzonase (2 µg/ml) and lysozyme (50 mg per 50 ml). After sonication for 10 min at high intensity (10"on/15"off) and 5 min at medium intensity (10"on/15"off) extracts were centrifuged for 20 min at 4 °C at 40,000 × g. Extracts were incubated with 50 µl of magnetic Glutathione (Thermo Fisher Scientific) or Protein A (Cytiva) beads (each with 2.5 ml of protein extract) at RT for 2 h. Beads were collected and washed six times with extraction buffer (without benzonase and lysozyme) and finally resuspended in 80 µl of 1 × SDS-PAGE loading buffer prepared in extraction buffer and boiled for 5 min. Proteins were then separated on an 12% SDS-PAGE gel and analyzed by either colloidal blue staining or Western blot. For SDS-PAGE, separation was performed using self-cast gels using standard methods. Gels were stained using colloidal blue reagents (MBS Blue) provided by the Molecular Biology Service. For Western blots, proteins were transferred using a standard wet protocol to a 0.2 µm nitrocellulose membrane (Cytiva Cat. No. 10600004), blocking in 5% non-fat dry milk (Maresi) dissolved in PBST. Anti-His (Sigma-Aldrich, H1029) and anti-GST (Santa Cruz Biotechnology, sc-138) antibodies were used at 1:1000 diluted in blocking buffer and detected using chemiluminescence following incubation with anti-mouse HRP conjugates (Biorad Cat. No. 1706516), which were diluted 1:10,000 in blocking buffer. Imaging of gels and Western blots was performed using a Thermo Fisher iBright 1500 imaging system.

## In vivo co-immunoprecipitation experiments

Endogenous co-immunoprecipitation (co-IP) in *S. pombe* was performed following a previously published protocol [130] with minor modifications. Briefly, 0.5 g of cell pellets per strain were lysed using acid-washed glass beads in a Precellys for 4x20 sec maximum speed, 1 min on ice between rounds in ice-cold 1 × lysis buffer (50 mM Na-HEPES pH 7.5, 200 mM NaOAc, 1 mM EDTA, 1 mM EGTA, 5 mM MgOAc, 5% glycerol, 0.25% NP-40, 3 mM DTT, Roche EDTA-free protease inhibitors). Lysates were cleared at 20,000 × g for 5 min. Next, 40 µl of anti-FLAG M2 agarose beads were pre-washed and incubated with 500 µl normalized lysate for 2 h at 4°C with end-over-end rotation. Beads were then washed 4× with ice-cold wash buffer (same as lysis buffer), then eluted in 1 × SDS sample buffer at 95°C for 5 min. Inputs (10%) and IPs were then separated on an 8% SDS-PAGE gel and analyzed by Western blot. For SDS-PAGE, separation was performed using self-cast gels using standard methods. Proteins were transferred using a standard wet protocol to a 0.2 µm nitrocellulose membrane (Cytiva Cat. No. 10600004), blocking in 5% non-fat dry milk (Maresi) dissolved in PBST. Anti-FLAG M2

(Sigma-Aldrich; F3165) or anti-myc (IMP Molecular Biology Service, AB_558470) antibodies were used at 1:1000 dilution in blocking buffer and detected using chemiluminescence following incubation with anti-mouse HRP conjugates (Biorad Cat. No. 1706516), which were diluted 1:10,000 in blocking buffer. Imaging of gels and Western blots was performed using a Thermo Fisher iBright 1500 imaging system.

### mRNA sequencing library preparation and analysis

Total RNA was extracted from yeast strains grown to mid-exponential phase ($OD_{600}$ ~ 0.5) using a standard hot acid phenol protocol. Libraries were then prepared by polyA-tail enrichment using the NEBNext Ultra II Directional RNA Library Prep Kit for Illumina (NEB Cat. No. E7760L) with the polyA selection module (NEB Cat. No. E7490L). Size selection steps and final library cleanup were performed using the NA clean-up bead solution provided by the VBC core facilities, which is adapted from DeAngelis et al. 1995 [131] using carboxylate-modified Sera-Mag Speed beads (Cytiva). Library quality control was performed using the Advanced Analytical Fragment Analyzer using the HS NGS fragment analysis kit (Agilent Cat. No. DNF-474) as well as qPCR using reagents produced by the Molecular Biology Service in conjunction with commercial DNA standards (Roche Cat. No KK4903). Data was collected on a Thermo Scientific QuantStudio5 instrument.

Prepared libraries were sequenced on either an Illumina NextSeq2000 or NovaSeq S4 at the Vienna BioCenter Core Facilities Next Generation Sequencing Core using paired-end mode. Quality control of the data was performed using FastQC (Babraham Institute). Transcript-level quantification against the *S. pombe* ASM*294v2 reference genome* available from ENSEMBLv55 was performed using Kallisto [132]. Differential expression analysis was performed using DESeq2 [133] in R, where all subsequent data manipulation was performed.

### PRO-seq library generation and analysis

We performed a variant of PRO-seq, qPRO-seq, as recently published [73], with modifications [134] to make it compatible with *S. pombe*. *S. pombe* cells were grown in 10 ml YES media to $OD_{600}$ ~ 0.4–0.5. Cells were then harvested by centrifugation at 400 x *g* for 5 min at 4 ºC. Cells were then resuspended in 10 ml of ice-cold PBS, spun down once more, then resuspended in 10 ml ice-cold yeast permeabilization buffer (0.5% Sarkosyl, 0.5 mM DTT, Roche cOmplete protease inhibitor cocktail, 4U/ml Invitrogen RiboLock RNAse Inhibitor). After 20 min incubation on ice, cells were once again collected by gentle centrifugation, then resuspended in 50 µl storage buffer (10 mM Tris-HCl, pH 8.0, 25% (v/v) glycerol, 5 mM $MgCl_2$, 0.1 mM EDTA, 5 mM DTT).

A 2X run-on reaction master mix (40 mM Tris pH 7.7, 64 mM MgCl2, 1 mM DTT, 400 mM KCl, 40 µM Biotin-11-CTP, 40 µM Biotin-11-UTP, 40 µM ATP, 40 µM GTP, 1% sarkosyl) was prepared and preheated to 30 ºC. To each aliquot of permeabilized cells, 50 µl of master mix and 1 µl of RNAse inhibitor were added, then mixed by gently pipetting with wide-bore pipet tips before immediately incubating at 30 ºC for 5 min. To ensure precise timing, samples were done in batches staggered by 30–60 sec. Using the Norgen RNA Extraction Kit (Cat. No. 37500), 350 µL of RL buffer were added to each sample and then vortexed. 240 µl of absolute ethanol were added, then the entire well-mixed solution was applied to the RNA extraction column. The column was spun at 3500 x *g* for 1 min at 25 ºC, then washed once with 400 µl of wash solution A. This step was repeated a second time, then the column was dried by spinning at high speed for 2 min. RNA was eluted twice in 50 µl of DEPC-treated MonoQ water, the two fractions were pooled and then briefly denatured at 65 ºC for 30 sec. After snap cooling on ice, 25 µl of ice cold 1 M NaOH were added to each sample, which was then incubated for 10 min on ice. Finally, samples were quenched with 125 µl of cold 1 M Tris-HCl pH 6.8, then cleaned up using a standard ethanol precipitation overnight. Samples were resuspended in a small volume of DEPC water (6 µl).

Meanwhile, 10 µl per sample of Dynabeads MyOne Streptavidin C1 beads (Thermo Fisher Cat. No. 65001) were prepared by washing once with bead preparation buffer (0.1 M NaOH, 50 mM NaCl), then once in bead binding buffer (10 mM Tris-HCl, pH 7.4, 300 mM NaCl, 0.1% (v/v) Triton X-100, 1 mM EDTA, RNAse inhibitors), then resuspended in 25 µl bead binding buffer.

3' RNA adapter ligation was performed by adding 1 µl 10 µM oligo VRA3 to the resuspended RNA, boiling at 65 ºC for 30 sec, then snap cooling on ice. Ligation reactions were then prepared with T4 RNA Ligase using the provided instructions. After incubating reactions at 25 ºC for 1 hr, ligated RNA was bound to the 25 µl previously prepared streptavidin beads by adding 55 µl bead binding buffer and the entire ligation reaction. Samples were incubated for 20 min at 25 ºC, then washed once with 500 µl high salt wash buffer (50 mM Tris-HCl, pH 7.4, 2 M NaCl, 0.5% (v/v) Triton X-100, 1 mM EDTA, 2 µl/10 ml Ribolock RNAse inhibitor), transferring beads to a clean tube. Beads were then washed once more in the new tube with low salt wash buffer (5 mM Tris-HCl, pH 7.4, 0.1% (v/v) Triton X-100, 1 mM EDTA, 2 µl/10 ml Ribolock RNAse inhibitor), then resuspended in 20 µl T4 PNK reaction mix (1X T4 PNK Buffer, 1 mM ATP, 1 µl T4 PNK, 1 µl Ribo-Lock RNAse inhibitor). Samples were then incubate at 37 ºC for 30 min, then the beads were collected by magnet and resuspended in 20 µl ThermoPol reaction mix (1X ThermoPol buffer, 1 µl RppH, 1 µl RiboLock RNAse inhibitor), then incubated at 37 ºC for 1 hr. Beads were once again collected by magnet, then the 5' RNA adaptor (REV5) was ligated using T4 RNA ligase, setting up the reaction using the provided manual and incubating at 25 ºC for 1 hr. Beads were then washed once with high salt wash buffer, transferring to a clean tube, then once with low salt buffer, before being resuspended in 300 µl of TRIzol reagent (Ambion). Samples were vortexed for 20 sec, then incubated on ice for 3 min before adding 60 µl of chloroform, vortexing for 15 sec, then incubating on ice for a further 3 min. Samples were then spun at 20,000 x $g$ for 8 min at 4 ºC, then the upper ~180 µl aqueous phase was transferred to a clean tube and precipitated using a standard ethanol precipitation.

Finally, pellets were resuspended in 13.5 µl reverse transcription resuspension mix (8.5 µl DEPC water, 4 µl 10 µM primer RP1, 1 µl 10 mM dNTPs), denaturing at 65 ºC for 5 min, snap cooling on ice. Reverse transcription was then performed using Maxima H Minus RT (Thermo Fisher Cat. No. EP0753) by adding 6.5 µl RT master mix (4 µl 5X RT buffer, 1 µl DEPC water, 0.5 µl RiboLick RNAse inhibitor, 1 µl Maxima H Minus RT), then cycling as follows: 50 ºC for 30 min, 65 ºC for 15 min, 85 ºC for 5 min. 2.5 µl of indexed RPI-n primer (10 µM) were then added to each RT reaction, then PCR was performed as a 100 µl reaction using the Q5 polymerase (NEB) with the included high GC content enhancer. Samples were then PCR-amplified to the beginning of the logarithmic phase, as determined using a small-scale qPCR reaction run in parallel (typically less than 20 cycles).

Libraries were sequenced as described previously at the Vienna BioCenter Core Facilities Next Generation Sequencing Facility. The data analysis was performed by adapting available pipelines [73]. Briefly, we created a Nextflow pipeline (https://github.com/Gregor-Mendel-Institute/PROalign) to perform all QC, alignment, and read processing steps, generating strand-specific bigwig files that were subsequently analyzed in R.

**Long-read transcriptomics (PacBio Iso-seq)**

Total RNA was extracted from yeast strains grown to mid-exponential phase (OD600 ~ 0.5) using a standard hot acid phenol protocol. For PacBio Iso-Seq, the sequencing libraries were prepared by DNA Link Sequencing Labs with the SMRT-bell Barcoded Adapter Plate 3.0, followed by Iso-Seq sequencing on the Revio platform using SMRT Link v25.1. CCS reads (also known as HiFi reads) and high-quality isoforms were identified using the Iso-Seq4 (https://github.com/Pacific-Biosciences/IsoSeq) analytic software with a default parameter. This approach yielded over 15 million high-fidelity (HiFi) reads (Q ≥ 20) with a median read length of 1,702 bp, closely matching the median transcript length in *S. pombe* (1,883 bp). Subsequent analyses were performed in-house as follows: the clustered transcripts were mapped to the *Schizosaccharomyces pombe* genome using minimap2 v2.24 [135] with the parameters "-ax splice:hq -uf". Redundant or highly similar transcripts were collapsed using TAMA Collapse with parameter "-a 100 –m 100 –x no_cap". Collapsed transcript sets from the two independent libraries were subsequently merged using TAMA Merge [136] with parameter "-d merge_dup".

To identify antisense transcripts, we downloaded the GFF3 file containing gene annotations from the Ensembl FTP (https://ftp.ensemblgenomes.ebi.ac.uk/pub/fungi/release-60/gff3/schizosaccharomyces_pombe/). Features annotated as "antisense" and/or "non-coding" were excluded. The resulting reduced GFF3 file was then used as a reference for

gffcompare [137]. Long-read transcripts classified by gffcompare as "exonic overlap on the opposite strand" (class code 'x') were directly adopted as *our* antisense transcript annotation.

## MNase-seq library generation and analysis

We performed MNase-seq by adapting a published protocol [138]. *S. pombe* strains (100 ml) were grown to mid-exponential phase ($OD_{600}$ ~ 0.4-0.5) in YES, fixed with 0.5% formaldehyde for 20 min at RT, quenched with 125 mM glycine for 10 min, then collected by centrifugation. Cells were then washed with 10 ml $dH_2O$, followed by resuspension into 5 ml of ice cold preincubation solution (20 mM citric acid, 20 mM $Na_2HPO_4$, 40 mM EDTA pH 8.0, 30 mM b-mercaptoethanol), and incubation in a 30 ºC water bath for 10 min. Cells were next harvested and resuspended in 2 ml Sorbitol/Tris buffer (1 M sorbitol, 50 mM Tris/HCl pH 7.5, 10 mM b-mercaptoethanol), and 2 mg of Zymolyase 100T lyophilized powder (Nacalai Tesque) was resuspended in 40ul of water and added per 100 ml culture to produce spheroplasts. Samples were then incubated in a 30 ºC water bath for 30 min, before harvesting and washing with 10 ml Sorbitol/Tris buffer (without b-mercaptoethanol). The spheroplasts were then resuspended in 1 ml of ice-cold NP buffer (1 M sorbitol, 50 mM NaCl, 10 mM Tris/HCl pH 7.5, 5 mM $MgCl_2$, 1 mM $CaCl_2$, 0.75% NP-40, 1 mM b-mercaptoethanol, 0.5 mM spermidine).

For MNase digestion, 1.5 U of MNase (Sigma-Aldrich) was added to each sample, and samples were incubated at 37 ºC for 15 min. The reaction was stopped by the addition of 138 µl of Stop Buffer (5% SDS, 100 mM EDTA). Next, 40 µl of 10 mg/ml RNAse A (Thermo Fisher) was added and samples were incubated at 37 ºC for 45 min, followed by the addition of 50 µl of 5 mg/ml Proteinase K (Thermo Fisher) and further incubation at 65 ºC overnight. The next morning, 360 µl of potassium acetate pH 5.5 was added to each sample, followed by incubation on ice for 10 min. DNA was then isolated from the supernatant using classical phenol-chloroform-isopropanol extraction and ethanol precipitation methods. Extracted DNA was then run on a 2% agarose gel, and bands corresponding to mononucleosomal DNA was recovered by gel extraction. DNA concentration was measured by Nanodrop spectrophotometer.

For library preparations, all samples, were processed using the NEBNext Ultra II library preparation kit for Illumina (NEB Cat. No. E7645L) using custom synthesized barcoded sequencing adapters provided by the Vienna BioCenter Core Facilities Next Generation Sequencing Facility. Following quality control by Agilent Fragment Analyzer and quantification by RT-qPCR using a Kapa Library Quantification Kit (Roche Cat. No. KK4903), libraries were sequenced at the Next Generation Sequencing Facility on an Illumina NovaSeq X. Alignments, quality control, and initial processing were performed using the nf-core/mnaseseq pipeline (https://github.com/nf-core/mnaseseq) with default parameters. Further analysis was performed using Deeptools (v3.3.1) [139].

For the calling of nucleosome dyad positions, we used the Numap package [140] (https://github.com/orphancode/NuMap, https://github.com/dblyons/MNase_seq) with the call_dyads function on default parameters. For each dyad, ± 50 bp windows (100 bp) were extracted with bedtools (v2.27.1) getfasta. GC% was computed as (G + C)/(A + C + G + T) × 100 using Python (v3.6.6). Distributions and statistical tests were plotted in R (v4.3.1).

## ChIP-seq library generation and analysis

*S. pombe* strains (100 ml) were grown to mid-exponential phase ($OD_{600}$ ~ 0.4-0.5) in YES, fixed with 1% formaldehyde for 10 min, quenched with 125 mM glycine for 10 min, then collected by centrifugation. Cells were washed once with ice-cold PBS, then resuspended in lysis buffer (50 mM HEPES-KOH, pH 7.5, 140 mM NaCl, 1 mM EDTA, 0.1% sodium deoxycholate, 1% Triton X-100, 1X Roche Complete Protease Inhibitors), and then lysed using acid-washed glass beads in a Precellys for 4x20 sec maximum speed, 1 min on ice between rounds. Lysate was then separated from the beads and then chromatin was sheared to a predominant final DNA length of ~150–200 bp in a Covaris E220 ultrasonicator, and then clarified of unlysed cells and debris by centrifugation twice at 16,000 × g for 10 min. Prepared chromatin was then frozen at -70 ºC pending use.

Chromatin aliquots were thawed on ice, then normalized based on starting culture density using additional lysis buffer for dilution. 10% of normalized and diluted chromatin were reserved as input controls. Either anti-Myc 9E10 (1μg; IMP Molecular Biology Service, AB_558470), anti-H3K36me3 (1μg; Abcam, ab9050), anti-H2BK120ub (1μg; Active Motif, Cat#39623) or anti-H2A.Z (2μg; Active Motif, Cat#39640) antibodies were then added to each culture and samples were incubated overnight at 4 ºC with inversion. 25 μl of Protein A Dynabeads (Thermo Fisher Cat. No. 10001D) per sample were prepared by washing twice with PBS with 0.1% tween-20, once with lysis buffer, then resuspending in 100 μl of lysis buffer. Prepared beads were then combined with overnight chromatin samples and incubated for 4 hrs at 4 ºC with inversion. Beads were then collected using the magnet and washed twice with lysis buffer, twice with high salt lysis buffer (50 mM HEPES-KOH, pH 7.5, 500 mM NaCl, 1 mM EDTA, 0.1% sodium deoxycholate, 1% Triton X-100), twice using lithium wash buffer (10 mM Tris-HCl, pH 8, 250 mM LiCl, 1 mM EDTA, 0.5% NP-40, 0.5% sodium deoxycholate), and once with TE. Beads were then resuspended in 100 μl of elution buffer (50 mM Tris-HCl, pH 8, 10 mM EDTA, 0.8% SDS) and transferred to PCR strips. Samples were then boiled for 10 min at 95 ºC, then 65 ºC overnight in a thermocycler. The next morning, proteinase K was added (final concentration 0.2 mg/mL) and incubated at 55 °C for 2 h. Samples were then cleaned up using a commercial kit (Zymo Research Cat. No. D5201). DNA concentration was measured by Nanodrop spectrophotometer.

For library preparations, all samples, including input, were processed using the NEBNext Ultra II library preparation kit for Illumina (NEB Cat. No. E7645L) using custom synthesized barcoded sequencing adapters provided by the Vienna BioCenter Core Facilities Next Generation Sequencing Facility. Following quality control by Agilent Fragment Analyzer and quantification by RT-qPCR using a Kapa Library Quantification Kit (Roche Cat. No. KK4903), libraries were sequenced at the Next Generation Sequencing Facility on an Illumina NovaSeq X. Alignments, quality control, and initial processing were performed using the nf-core/chipseq pipeline (https://github.com/nf-core/chipseq) with default parameters. Input-normalized bigWig files were subsequently generated using Deeptools (v3.3.1) [139].

### Promoter and motif analyses

For all promoter analyses, we defined 500 bp regions upstream of sense and antisense TSSs as promoter regions. The TATA position weight matrix (PWM) was taken from the JASPAR CORE Fungi (redundant v2) database. We scanned promoters with FIMO (https://meme-suite.org/; $P < 1e-4$) to count TATA occurrences and summarize positions relative to TSS. We used STREME for de novo motif discovery on antisense promoters (default parameters; $E \leq 0.05$). Motifs (including TATA) were matched to *S. pombe* TF motifs using TOMTOM against a published TF motif atlas [79] (default parameters; $FDR \leq 0.05$).

### AlphaFold predictions

We ran either AlphaFold-Multimer v2.2 or AlphaFold3 [141] (https://alphafoldserver.com/) for all protein complex predictions. Of the five models generated, we used only the model ranked 1 for further processing and interpretation. We used the software UCSF ChimeraX (v1.8) [142] for the visualization and superimposition of AlphaFold models. For the *in-silico* protein interaction screen, we used a previously established pipeline [143] that integrated MMseqs and ColabFold [144] (https://gitlab.com/BrenneckeLab/ht-colabfold).

### Multiple sequence alignments

Multiple sequence alignments were conducted for homologs of Hrp3 and Prf1 across 11 representative eukaryotic model species: *Schizosaccharomyces pombe, Saccharomyces cerevisiae, Neurospora crassa, Caenorhabditis elegans, Drosophila melanogaster, Danio rerio, Mus musculus, Homo sapiens, Physcomitrella patens, Arabidopsis thaliana,* and *Oryza sativa*. Alignments were performed using the Multiple Alignment using Fast Fourier Transform (MAFFT) [145] algorithm.

Homologs were identified as proteins clustering within the same orthogroup, based on Orthofinder [146] (version 2.5.4) analysis performed with default settings.

**Quantification and statistical analysis**

Unless otherwise noted, all statistical analyses were performed in R (v4.1.3) using the RStudio IDE (v2022.12.0 + 353) on x86_64-apple-darwin17.0 (64-bit) under macOS 13.6.3. All statistical tests used are stated in corresponding figure legends. Processing of NGS data, as well as all AlphaFold2 Multimer analyses, were performed using the CLIP cluster (https://clip.science).

## Supporting information

**S1 Fig. H2A.Z Enrichment in 11 Nucleosome Remodeler Mutants in Fission Yeast.**
(DOCX)

**S2 Fig. Analyses of Sense and Antisense Expression in *hrp1Δ*, *hrp3Δ* and *hrp1Δhrp3Δ*.**
(DOCX)

**S3 Fig. Additional Analyses of Antisense Transcripts in *hrp1Δ*, *hrp3Δ* and *hrp1Δhrp3Δ*.**
(DOCX)

**S4 Fig. Loss of *hrp3* is Associated with Depletion of Nucleosome Occupancy over AT-rich Regions.**
(DOCX)

**S5 Fig. Characteristics of Cryptic Antisense Promoters Identified in *hrp3Δ* and *hrp1Δhrp3Δ*.**
(DOCX)

**S6 Fig. Further Analyses on MUGs and Convergent Nested Genes in Fission Yeast.**
(DOCX)

**S7 Fig. Comparative Domain Analysis of Hrp1 and Hrp3.**
(DOCX)

**S8 Fig. AlphaFold-predicted Interactions Between Prf1 and the Hrp3 CHCT Domain Across Species.**
(DOCX)

**S9 Fig. Interacting Residues Between Hrp3 and Prf1 are Conserved Across Eukaryotes.**
(DOCX)

**S10 Fig. Additional Analyses of *prf1* Mutants.**
(DOCX)

**S11 Fig. Analysis of cryptic sense transcription in *hrp1Δhrp3Δ*.**
(DOCX)

**S1 Note. Transcription of cryptic sense isoforms in *hrp1Δhrp3Δ*.**
(DOCX)

**S1 Table. *De novo* motif discovery at cryptic antisense promoters.**
(DOCX)

**S2 Table. Early, middle and late MUGs upregulated in *hrp3Δ*.**
(DOCX)

**S3 Table. Top candidates for predicted interaction with the CHCT domain of Hrp3.**
(DOCX)

**S4 Table. *Schizosaccharomyces pombe* strains used in this study.**
(DOCX)

## Acknowledgments

We thank the entire Berger group, for their insightful, considerate, and helpful discussions. We particularly thank Pierre Bourguet, Chung Hyun Cho and Zdravko Lorković for their invaluable feedback on the manuscript. We are especially grateful to Hwan Su Yoon (Sungkyunkwan University, Korea) for their support with long-read transcriptomics and to Dominik Handler for providing scripts for AlphaFold. We also acknowledge the Molecular Biology Service and Media Kitchen for their supply of plates, basic reagents, and sequencing services. We also thank the Vienna BioCenter Core Facilities, in particular the Next Generation Sequencing and Protein Technologies facilities for their advice and handling of all our requests and helpful discussions. For open access purposes, the authors have applied a CC BY public copyright license to any author accepted manuscript version arising from this submission.

## Author contributions

**Conceptualization:** Frederic Berger.

**Data curation:** Jian Yi Kok.

**Formal analysis:** Jian Yi Kok.

**Funding acquisition:** Frederic Berger.

**Investigation:** Jian Yi Kok.

**Methodology:** Jian Yi Kok, Zachary H Harvey.

**Project administration:** Frederic Berger.

**Resources:** Jian Yi Kok, Elin Axelsson.

**Software:** Elin Axelsson.

**Supervision:** Zachary H Harvey, Elin Axelsson, Frederic Berger.

**Validation:** Jian Yi Kok, Zachary H Harvey, Elin Axelsson.

**Visualization:** Jian Yi Kok.

**Writing – original draft:** Jian Yi Kok, Frederic Berger.

**Writing – review & editing:** Jian Yi Kok, Zachary H Harvey, Elin Axelsson, Frederic Berger.

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
