## [Decision Letter · Decision Letter 0]

22 Jan 2026

PGENETICS-D-25-01380

Nucleosome Positioning Shapes Cryptic Antisense Transcription

PLOS Genetics

Dear Dr. Berger,

Thank you for submitting your manuscript to PLOS Genetics. After careful consideration, we feel that it has merit but does not fully meet PLOS Genetics's publication criteria as it currently stands. Therefore, we invite you to submit a revised version of the manuscript that addresses the points raised during the review process.

Please submit your revised manuscript within by Feb 21 2026 11:59PM. If you will need more time than this to complete your revisions, please reply to this message or contact the journal office at plosgenetics@plos.org. Please include the following items when submitting your revised manuscript:

We look forward to receiving your revised manuscript.

Kind regards,

Monica P. Colaiácovo

Section Editor

PLOS Genetics

Aimée Dudley

Editor-in-Chief

PLOS Genetics

Anne Goriely

Editor-in-Chief

PLOS Genetics

**Additional Editor Comments:**

In particular, please address the point raised by both Reviewers 1 and 3 regarding the co-IP using Prf1-Flag, and the missing control and normalization for the qPCR assay (Reviewer 3). Most other comments can be addressed by either inserting already available information into the main text or adjusting/modifying the text accordingly (Points 5 and 11, respectively, for Reviewer 3).

**Journal Requirements:**

https://journals.plos.org/plosgenetics/s/submission-guidelines#loc-parts-of-a-submission

3) Your manuscript is missing the following sections: Abstract.  Please ensure all required sections are present and in the correct order. Make sure section heading levels are clearly indicated in the manuscript text, and limit sub-sections to 3 heading levels. An outline of the required sections can be consulted in our submission guidelines here:

https://journals.plos.org/plosgenetics/s/submission-guidelines#loc-parts-of-a-submission

6)  Thank you for including the following in your data availability statement: 'https://github.com/Gregor-Mendel-Institute/kok_2025/tree/main/output/annos.'  However, the accession number provided is currently inaccessible.

7) Please verify and provide an active link to facilitate access to the data.

Please ensure that the funders and grant numbers match between the Financial Disclosure field and the Funding Information tab in your submission form. Note that the funders must be provided in the same order in both places as well.

**Reviewers' comments:**

Reviewer's Responses to Questions

**Comments to the Authors:**

Reviewer #1: Kok et. al. adequately addressed and clearly responded to all but one point raised and I would like to recommend publication of their work once the point below is addressed.

The only experiment requiring further clarification is the co-immunoprecipitation followed by western blotting to demonstrate the Hrp3–Prf1 interaction in vivo (Reviewer 3, Major Comment 3; Reviewer 2, Comment 12). To interpret this experiment, it is necessary to include a FLAG pulldown followed by anti-Myc western blot in cells where Prf1 is not FLAG-tagged. Given the weak interaction observed, background/non-specific signal from Hrp3 is a technical concern. Depending on the results, the authors should either include this control or adjust their claims regarding an in vivo interaction accordingly.

Reviewer #2: The authors have thoroughly addressed all of my comments. The revised manuscript is clear, rigorous, and represents a high-quality study that will be of significant interest to the field.

Reviewer #3: Kok et al. have addressed all my comments and suggestions on their manuscript very thoroughly. However, I still have some additional comments on some of the new experiments.

Point 5.

The explanations of Figures 1D, 2B and 3A and 3B in the two paragraphs are clear and well described. I think these explanations, or a summary of them, should be included in the text to prevent readers from having similar doubts to those I had when reading the manuscript.

Point 10.

The negative control is still absent. The new qPCR assay histogram should show eight columns: the mRNA level of the wild-type ura4 gene (positive control), the level in the strain where its promoter has been replaced by that of adh1, the level in the strain where the ura4 promoter has been deleted (negative control), and the level of the five putative promoters to be tested. The results should be presented by normalization of the ura4 level in each strain relative to act1 without the need to refer them to the strain where the ura4 promoter has been deleted. This would enable the promoter efficiency in the different strains to be compared directly.

Point 11.

I agree that a detailed analysis of the role of hrp1 and hrp3 in meiosis is beyond the scope of this work. However, without further analysis, it is impossible to determine whether defects in meiosis are caused by antisense-mediated deregulation of meiosis-specific genes, given that these genes only account for a fraction of all genes that are deregulated in mutants. Furthermore, this possibility is inconsistent with the observation that the hrp1Δ mutant exhibits more severe alterations to meiosis despite displaying significantly lower levels of antisense expression than the hrp3Δ mutant (Fig 1D and 2A).

Point 12.

In vivo co-immunoprecipitation experiments show that Hrp3-Myc is faintly detected after Prf1-FLAG immunoprecipitation. This experiment, however, lacks a negative control using either agarose beads or agarose beads coupled to an unrelated antibody, which would show that the a-Myc signal is specific and not due to background. To provide further confirmation, the authors could carry out reverse co-immunoprecipitation, first with Hrp3-Myc and then with Prf1-FLAG.

Point 13.

The new experiment in liquid medium shows that after 15 hours of growth, hrp1∆ hrp3∆ prf1∆ cells reach a lower cell density than single prf1∆ or double hrp1∆ hrp3∆ mutants. The authors should measure and represent the optical density (OD) at regular intervals over a 15-hour period, starting from an identical initial OD. This straightforward experiment would yield more detailed information about the growth rates of the four strains.

**Have all data underlying the figures and results presented in the manuscript been provided?**

Large-scale datasets should be made available via a public repository as described in the *PLOS Genetics*
data availability policy , and numerical data that underlies graphs or summary statistics should be provided in spreadsheet form as supporting information., and numerical data that underlies graphs or summary statistics should be provided in spreadsheet form as supporting information.

Reviewer #1: Yes

Reviewer #2: Yes

Reviewer #3: None

PLOS authors have the option to publish the peer review history of their article (what does this mean? ). If published, this will include your full peer review and any attached files.). If published, this will include your full peer review and any attached files.

**Do you want your identity to be public for this peer review?** For information about this choice, including consent withdrawal, please see our For information about this choice, including consent withdrawal, please see our Privacy Policy ..

Reviewer #1: **Yes:** Jakob Schnabl-BaumgartnerJakob Schnabl-Baumgartner

Reviewer #2: No

Reviewer #3: No

**Figure resubmission:**
---

## [Editor Report · Decision Letter 1]

2 Mar 2026

Dear Dr Berger,

We are pleased to inform you that your manuscript entitled "Nucleosome Positioning Shapes Cryptic Antisense Transcription" has been editorially accepted for publication in PLOS Genetics. Congratulations!

Yours sincerely,

Monica P. Colaiácovo

Section Editor

PLOS Genetics

Aimée Dudley

Editor-in-Chief

PLOS Genetics

Anne Goriely

Editor-in-Chief

PLOS Genetics

BlueSky: @plos.bsky.social

Comments from the reviewers (if applicable):

**Data Deposition**

If you have submitted a Research Article or Front Matter that has associated data that are not suitable for deposition in a subject-specific public repository (such as GenBank or ArrayExpress), one way to make that data available is to deposit it in the Dryad Digital Repository . As you may recall, we ask all authors to agree to make data available; this is one way to achieve that. A full list of recommended repositories can be found on our . As you may recall, we ask all authors to agree to make data available; this is one way to achieve that. A full list of recommended repositories can be found on our website ..

http://datadryad.org/submit?journalID=pgenetics&manu=PGENETICS-D-25-01380R1

Additionally, please be aware that our data availability policy  requires that all numerical data underlying display items are included with the submission, and you will need to provide this before we can formally accept your manuscript, if not already present. requires that all numerical data underlying display items are included with the submission, and you will need to provide this before we can formally accept your manuscript, if not already present.

**Press Queries**

If you or your institution will be preparing press materials for this manuscript, or if you need to know your paper's publication date for media purposes, please inform the journal staff as soon as possible so that your submission can be scheduled accordingly. Your manuscript will remain under a strict press embargo until the publication date and time. This means an early version of your manuscript will not be published ahead of your final version. PLOS Genetics may also choose to issue a press release for your article. If there's anything the journal should know or you'd like more information, please get in touch via plosgenetics@plos.org ..

---

## [Editor Report · Acceptance letter]

PGENETICS-D-25-01380R1

Nucleosome Positioning Shapes Cryptic Antisense Transcription

Dear Dr Berger,

We are pleased to inform you that your manuscript entitled "Nucleosome Positioning Shapes Cryptic Antisense Transcription" has been formally accepted for publication in PLOS Genetics! Your manuscript is now with our production department and you will be notified of the publication date in due course.

With kind regards,

Anita Estes

PLOS Genetics

On behalf of:
